# Malfunction of airway basal stem cells plays a crucial role in pathophysiology of tracheobronchopathia osteoplastica

Yue Hong [1✉], Shan Shan[2], Ye Gu[3], Haidong Huang[4], Quncheng Zhang[5], Yang Han[6], Yongpin Dong [7], Zeyu Liu[2], Moli Huang[8] & Tao Ren [2✉]

Understanding disease-associated stem cell abnormality has major clinical implications for prevention and treatment of human disorders, as well as for regenerative medicine. Here we report a multifaceted study on airway epithelial stem cells in Tracheobronchopathia Osteo-chondroplastica (TO), an under-detected tracheobronchial disorder of unknown etiology and lack of specific treatment. Epithelial squamous metaplasia and heterotopic bone formation with abnormal cartilage proliferation and calcium deposits are key pathological hallmarks of this disorder, but it is unknown whether they are coincident or share certain pathogenic mechanisms in common. By functional evaluation and genome-wide profiling at both transcriptional and epigenetic levels, we reveal a role of airway basal cells in TO progression by acting as a repository of inflammatory and TGFβ-BMP signals, which contributes to both epithelial metaplasia and mesenchymal osteo-chondrogenesis via extracellular signaling and matrix remodeling. Restoration of microenvironment by cell correction or local pathway intervention may provide therapeutic benefits.

[1] Stem Cell Center, Shanghai Jiao Tong University Affiliated Sixth People's Hospital, Shanghai, China. [2] Department of Respiratory Medicine, Shanghai Jiao Tong University Affiliated Sixth People's Hospital, Shanghai, China. [3] Department of Respiratory Medicine, Tongji University Affiliated Shanghai Pulmonary Hospital, Shanghai, China. [4] Department of Respiratory and Critical Care Medicine, Changhai Hospital, The Second Military Medical University, Shanghai, China. [5] Department of Respiratory Medicine, Henan Provincial People's Hospital, Zhengzhou, China. [6] Department of Pathology, Shanghai Jiao Tong University Affiliated Sixth People's Hospital, Shanghai, China. [7] Department of Emergency, Changzheng Hospital, The Second Military Medical University, Shanghai, China. [8] Department of Bioinformatics, School of Biology and Basic Medical Sciences, Soochow University, Suzhou, China. ✉email: hongyue2009@hotmail.com; rentao305@163.com

To evaluate and better understand the function and potential changes of diseases-associated tissue stem cells is a prerequisite and of great importance for a successful approach in autologous regenerative medicine. Epithelial stem cells, as key players in tissue homeostasis and injury-induced repair, are applied as a primary cell source in regenerative medicine[1–4]. In recent years, their participation in tissue remolding and disease progression has been increasingly implied, ranging from airway to intestinal chronic diseases, such as chronic rhinosinusitis, chronic obstructive pulmonary disease (COPD), idiopathic pulmonary fibrosis (IPF) and ulcerative colitis[5–8]. Adding to these pioneer findings, a lot more follow-up efforts are needed to improve the current knowledge in this aspect with an aim of covering a wider spectrum of pathophysiological conditions, especially those that are etiologically unknown and lack an effective treatment to date. Herein, we have extended our exploration of these factors on an idiopathic tracheobronchial abnormality, termed tracheobronchopathia osteochondroplastica (TO).

TO is a benign condition of the large airways firstly described in the 19th century, characterized by the presence of osseous and/or cartilaginous nodules in the submucosa of the tracheobronchial wall. During progression to a severe stage, the nodules protrude into the lumen of the trachea and main bronchi, which can lead to airway obstruction. Despite appearing as a rare disorder in the literature, TO is speculated to be more frequent than it has been reported with an incidence fluctuating from 1:125 to 1:6000 in bronchoscopy surveys[9,10]. Depending on the site and degree of tracheal/bronchial stenoses, it can be asymptomatic or present with non-specific respiratory symptoms (e.g. chronic cough, sputum production, hemoptysis, and dyspnea on exertion) for long before diagnosis and most patients were identified incidentally during intubation, computed tomography (CT) and bronchoscopy for other indications. To date, the underlying cause of TO remains unknown. Several theories have been proposed, including chronic airway inflammation, a congenital basis, chemical or mechanical irritation, metabolic disturbance, degenerative processes, ecchondrosis/exostosis, and metaplasia of elastic tissue[11–15], however, none of which has been firmly validated. There is no known genetic susceptibility to the development of this disease and no specific treatment currently available to prevent the formation of nodules. TO-associated recurrent infections and collapse of the lung are treated conventionally. For those who are at the early stage of the condition, inhaled corticosteroids may have some impact, but this is less beneficial for people with more advanced disease[16]. Thus, locating the key player of TO development and unveiling the potential mechanisms involved are of great clinical importance.

There are a few clues from bronchoscopic visualization and histopathological findings worthy of remark: (1) 48–78% of TO cases associated with the occurrence of squamous metaplasia of the tracheal epithelium[16–18], (2) prominent nodules formed in the portions where there is the absence of normal ciliated respiratory epithelium[19,20], and (3) the nodules are located along the airway cartilage side but sparing the posterior membranous wall of the trachea[21,22]. Furthermore, from developmental studies[23,24], it has been demonstrated that the epithelial and cartilage compartments are sharing a cross-talk machinery during embryonic mouse tracheal development and a loss of cartilage led to multiple epithelial phenotypes including a decrease in basal cell number. As basal cells are a well-recognized reservoir of stem cells responsible for epithelial homeostasis along conducting airways[25–27], taking all information above together, we speculated that there was a direct correlation of tracheal-bronchial basal cell (TBBC) with pathological changes on the epithelium and submucosal chondrogenesis and/or ossification, thereby resulting in

TO. This notion was preliminarily supported by our finding from culture later on (Supplementary Fig. 1a, b), that proliferative basal cells are more enriched in the anterior-lateral wall (cartilage portion) of human tracheae compared to the posterior membranous portion, which matches with the distribution of TO lesions.

In the present work, we cloned and functionally evaluated TBBCs from patients with TO via clonogenic assay, in vitro and in vivo differentiation, and epithelial–mesenchymal co-culture assays. By whole-genome expression (RNA-Seq) and epigenetic (assay for transposase-accessible chromatin, ATAC-seq) analysis, we provided a detailed disclosure of cellular alterations associated with this disease. Our results suggest that basal cells from TO epithelia are carrying sustained pathogenic changes and their malfunction may drive TO progression in both a local and interactive manner.

## Results

**Characterization of clonogenic cells in TO.** TBBCs were isolated from six patients with TO and seven non-TO donors with histologically normal upper airways via tracheal-bronchial brushing. The demographic characteristics of study subjects are shown in Table 1. Isolated basal cells were maintained and expanded conventionally on a lawn of irradiated 3T3-J2 (Fig. 1a, b), then proceeded with marker staining to confirm their stem cell signature. Both non-TO and TO cultures showed a uniform constituent of CK5+/p63+ clones which were active in proliferation (Ki67+) and free of differentiating cell markers, CK8, CK10, SCGB1A1, and Foxj1, regardless of passage number (Fig. 1c, Supplementary Fig. 2a, b). Despite many similarities, TO clones displayed diminished self-renewal compared to non-TO controls in terms of proliferation capacity reflected by the degree of Ki67 positivity and clonogenicity reflected by clone number (Fig. 1c, d). Taking advantage of a stable cell maintenance platform in vitro, on which disease-free TBBCs were kept clonogenic and multipotent-stable throughout long-term culture with just a mild decline of 10% in clonogenicity at Passage 20 (Supplementary Fig. 1c, d), a reducing trend of clonogenic ability was found on TO-derived cells over continuous passaging (Fig. 1d). Through an assay period of passage 3 to passage 10, the clonal frequency of TO-TBBCs was descending from 47% lower in TO versus Non-TO to more than 80% lower, whilst no significant difference appeared among passages on non-TO.

To further explore the differences at the transcriptional level, expression profiles of TBBCs from P3 to P4 cultures were studied via RNA-Seq for all donors (non-TO $n = 7$, TO $n = 6$). As a result, there were 512 differentially expressed genes (173 upregulated in non-TO, 339 upregulated in TO, FC > 2, $p$-value < 0.05) detected between two groups (Fig. 1e, Supplementary Table 1). Fifty-four top enriched genes are indicated (Fig. 1f). To look into details, genes related to keratinocyte differentiation (such as *CARD18*, *CERS3*, and *SPRR2* family), proinflammatory activity (such as *IL1A*, *IL32*, *S100A12*, and *TREM2*), osteochondro matrix disposition (such as *LTBP1*, *FN1*, and *COL1A1*) and injury-induced basal cell markers (such as *KRT14*, *KRT6B*, and *KRT6C*) were turned up in disease-derived TBBCs. In contrast, trachea specification and epithelial branching associated transcription factor (TF) Nkx2-1 and genes involved in cell cycle process and DNA replication expressed at higher levels in non-TO controls (Fig. 1f, g, Supplementary Fig. 3a). Furthermore, enrichment of pathways in EGFR signaling, interferon-γ signaling, and antimicrobial response in TO-derived TBBCs may result in possibly squamous-prone and goblet cell-prone fate determination, predicting a phenotypic change upon TBBC differentiation.

**Table 1 Demographics and clinical characteristics of study subjects.**

| Patient | Sex | Age (yr) | Smoking (Y/N) | Pathologic localization | Classification (type, stage) | Bronchoscopic image |
|---|---|---|---|---|---|---|
| TO-01 | M | 55–60 | N | Trachea | Scattered, I |  |
| TO-02 | M | 50–54 | – | Trachea, bilateral main bronchi, right upper bronchus | Diffuse, III |  |
| TO-03 | F | 55–60 | N | Trachea, bilateral main bronchi, right intermedius bronchus | Scattered, I |  |
| TO-04 | F | 70–74 | N | Trachea | Scattered, II |  |
| TO-05 | M | 50–54 | N | Trachea, bilateral main bronchi, right intermedius bronchus, left superior/ inferior bronchus | Confluent, III |  |
| TO-06 | M | 55–60 | Y | Trachea, right main bronchi | Diffuse, II |  |
| NonTO-01 | F | 35–40 | – | | | |
| NonTO-02 | M | 40–44 | N | | | |
| NonTO-03 | M | 65–70 | Y | | | |
| NonTO-04 | F | 70–74 | N | | | |
| NonTO-05 | M | 70–74 | N | | | |
| NonTO-06 | M | 25–30 | – | | | |
| NonTO-07 | F | 65–70 | N | | | |

| | Male sex, % | Average age (yr) |
|---|---|---|
| TO ($n = 6$) | 66.7 | 57.5 ± 8.3 |
| Non-TO control ($n = 7$) | 57.1 | 55.4 ± 19.0 |

– Information not available

In addition to the findings above, TO-TBBCs were shown to be dissimilar to control cells with respect to expression of additional basal cell markers NGFR/CK14[26,28] and cell morphology when cultured as monolayer under a feeder-free environment. NGFR expressed a higher heterogeneity in the p63+ population from TO cultures compared to its wide positivity across non-TO clones (Fig. 1h, Supplementary Fig. 2a, b). To be consistent, such a trend of reduction in NGFR expression was demonstrated by RNA-Seq data despite a p-value of 0.059 being reached in general. Stronger immunoreactivity of CK14 was observed on TO clones (Fig. 1i, Supplementary Fig. 2b). Taken into account that this is a putative injury-induced basal cell marker that elevates in response to two

circumstances- tissue injury and in vitro culture[28,29], the results obtained from comparisons between passage-matched non-TO and TO clones suggest the existence of additional injury memory in the TO-derived TBBCs.

Monolayer culture of TBBCs, a stringent condition for stem cell maintenance, indicated an obvious morphological difference between the two groups. While a uniformly compact p63+ cell layer formed by non-TO, TO cells appeared elongated and loose in arrangement to some extent, accompanying with fibronectin (FN1) expression (Fig. 1j, Supplementary Fig. 2c). FN1 is an important extracellular matrix component that plays a crucial role in skeletal system development and wound healing[30]. Besides its mesenchymal expression, this protein can be produced by epithelial cells when they expose to various stimuli such as

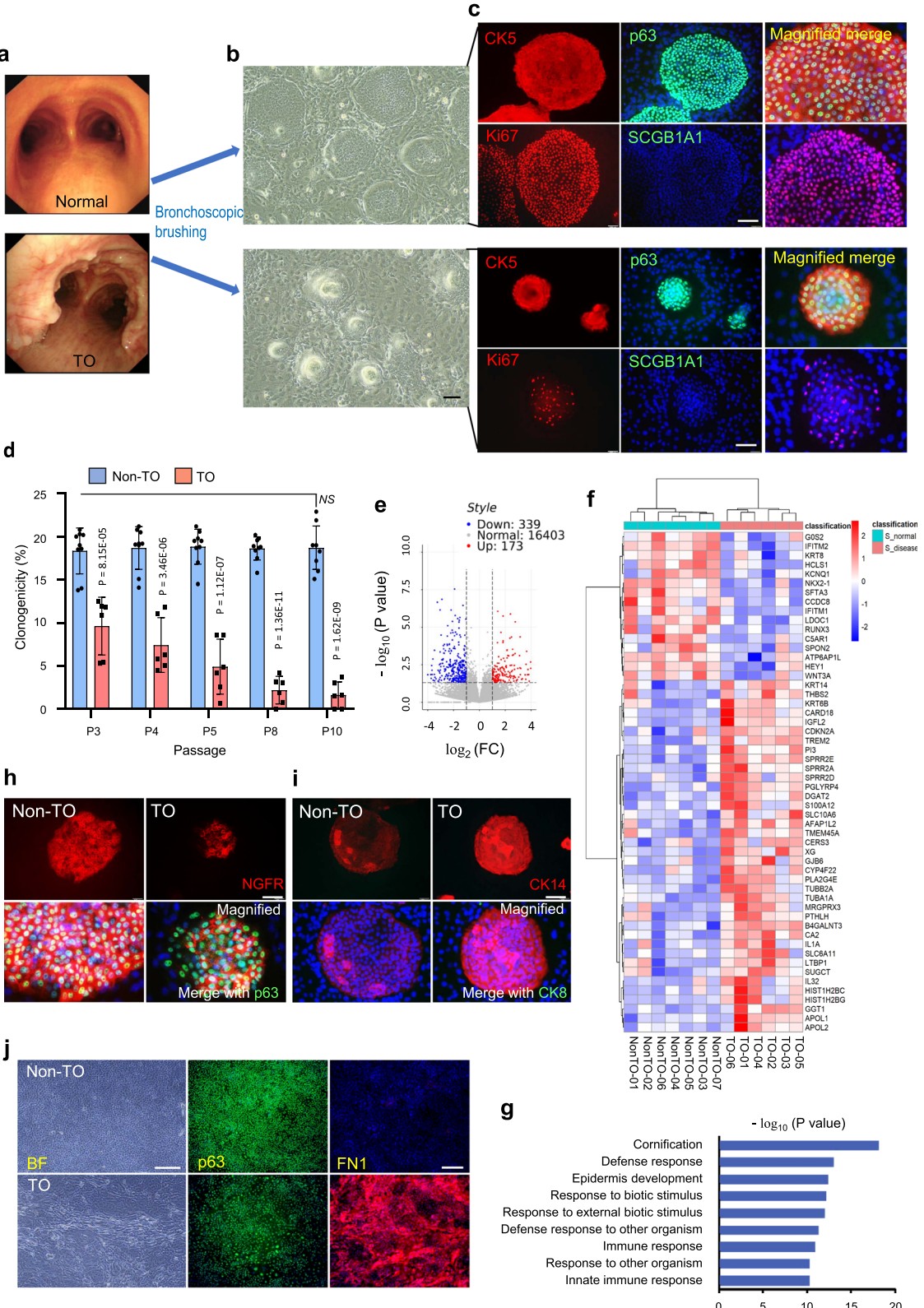

**Fig. 1 Tracheal-bronchial basal cell (TBBC) isolation and characterization. a** Bronchoscopic view of normal and typical TO influenced airway. **b** TBBCs cloned from brushing samples (non-TO $n = 7$, TO $n = 6$ biological independent samples) (**c**) were confirmed by immunofluorescent (IF) staining for basal cell signature markers p63/CK5, proliferation indicator Ki67, and exclusively checked for differentiating markers SCGB1A1. Scale bar, 100 μm. **d** Self-renewal ability was quantitatively evaluated, shown as percentage of plated cells that formed typical clones on Day 4–5 post-seeding over continuous passaging (non-TO $n = 9$ and TO $n = 6$ biologically independent experiments). Results are represented as mean ± SD, and $p$ values are indicated to show statistical significance based on two-tailed Student's $t$-tests. **e** Non-TO and TO-derived TBBCs from P3 to 4 culture were subjected to RNA-Seq. Volcano plot visualizing all genes according to the sequencing data, $p$-values ($-\log_{10}P$) plotted against fold changes ($\log_2$FC). Vertical dotted lines: $\log_2$FC = ±1, horizontal dotted lines: $-\log_{10}(P) = 1.30103$. $n = 6$–7/group. **f** Comparative expression profile was displayed by heatmap. **g** GO enrichment analysis of TO-derived TBBCs-upregulated genes. **h, i** p63, CK14, NGFR, and CK8 expression were assessed on cultured clones derived from control and TO cases. Scale bar, 100 μm. **j** Repsentative bright-field imaging and IF staining of fibronectin paralleled with p63 on TBBCs cultured on a feeder-free platform. Scale bar, 200 μm. **h, i, j** $n = 3$ biologically independent experiments. Source data are provided as a Source Data file.

hypoxia and tissue injury[31,32]. Taking all findings together, we speculated that TBBCs under TO conditions may already experience certain changes and carry a "memory" of cellular responses to the initial injury.

**Alterations of basal cell multipotency in TO.** Following the evaluations on cell identity and self-renewal ability, we next examined the self-commitment property of these disease-derived basal cells. Through in vitro differentiation on the air–liquid interface (ALI, 20-day culture), TO-TBBCs gave rise to a squamous-like structure with poor ciliogenesis, compared to a typical mucociliary differentiation that occurred under non-TO condition (Fig. 2a). There was no evidence of goblet cell hyperplasia detected on TO ALIs.

To further inspect the changes upon differentiation, ALI samples were subjected to RNA-sequencing for all donors. Corresponding to the striking difference observed at the structural level, distinct expression profiles (1266 genes upregulated, 1646 genes downregulated in TO versus non-TO, FC > 2, $p$-value < 0.05) and divergent differentiation paths were demonstrated (Fig. 2b, c). We found that TO cases were naturally grouped into two clusters in both whole-genome PCA and comparative heatmap. The cluster consisting of patients with stage I-II TO (TO-01, 03, 04), so labeled as "TO-moderate" in the following, is considered to present a progressive state, while the other consisting of stage II-III TO (TO-02, 05, 06), so labeled as "TO-severe", is representing the terminal changes. Comparing TO as a whole with non-TO controls, a robust enrichment of genes related to epithelial–mesenchymal transition, extracellular matrix constituent, angiogenesis, and the inflammatory response has been detected in the disease group, in contrast, genes involved in cilia biosynthesis and functioning were downregulated in concert under TO (Fig. 2d, e). Augmentation of TGFβ and EGFR signaling may provide an explanation for the structural change on TO ALIs[33–35].

To investigate further the difference between TO-moderate and TO-severe clusters, 442 out of 1266 TO-induced genes were differentially expressed and showed a progressive increment from TO-moderate to TO-severe (FC > 2). Through enrichment analysis, stronger hints in a number of aspects such as extracellular matrix organization, cytokine/chemokine signaling, and skeletal system development were demonstrated in the severe cluster (Supplementary Fig. 3b). Expression cascades of immune- and cartilaginous-axes were clearly turned up in the ALI samples derived from TO-TBBCs (Fig. 2f, Supplementary Fig. 4), suggesting an unneglectable role of epithelial elements in the TO development.

The self-commitment potential of patient-derived basal cells has been further evaluated for all donors via subcutaneous xenografting in immunodeficient NSG mice. Comparatively, an in vivo approach may behave more objectively owing to its nature of allowing spontaneous differentiation in a natural environment and supporting

observation of cellular interactions, whilst ALI assay benefits an easy collection of pure material for genomic or epigenetic analyses by maximally avoiding unwanted cell contamination. As a result, from xenografting, non-TO growths were usually larger in size and cyst-like, yet TO nodules were tiny and solid (Fig. 3a). Structures that originated from injected TBBCs were confirmed by staining with a human-specific nucleolar marker (Fig. 3b). Unlike the typical mucociliary pseudostratified epithelium formed by controls, TO-derived cells yielded a dysplasia phenotype with obvious squamous-like structures (Fig. 3c–e). Differentiation frequencies of specialized cell types including ciliated (Foxj1+), goblet (Muc5AC+) cells, and keratinocyte (IVL+) were compared between control and TO groups by quantifying immunohistochemistry-labeled xenografts (Fig. 3f). Again, here we observed increased expression of *RUNX2* and *SOX9* (master regulators of osteochondral specification) in TO xenografts versus controls (Fig. 3g). Despite no mature osteoblast or chondrocyte evidently visualized by marker staining on TO xenografts, tiny suspectable cartilaginous islets were occasionally seen in regions negative for human nuclei staining (Supplementary Fig. 5). This favors a hypothesis that TO epithelia may elicit excessive chondrogenesis and/or ossification in submucosa by orienting the mesenchyme, rather than basal cells themselves participating in *trans*-differentiation into bony/cartilaginous tissue.

**Augmented BMP-Smad signaling in TO-derived epithelia.** Bone morphogenetic protein-2 (BMP2), a member of the TGFβ superfamily, is one of the main chondrogenic growth factors involved in skeletal tissue regeneration and has been reported to express in mesenchymal cells and chondroblasts lining the nodules in TO[36]. As shown by our RNA-Seq data, BMPs and downstream targets (such as *SOX9* and *MSX2*) were elevated in ALI epithelia derived from TO-TBBCs. With a consideration that TBBCs have undergone at least 40-day culture in vitro through a course of expansion and ALI differentiation in the absence of an original disease microenvironment, we thereby, suspected that there may be a feedback loop self-promoting BMP production and signaling in TO epithelia originated from basal cells. To address this, we examined BMP2 expression and phospho-Smad levels on TO biopsies (donor $n = 3$, control $n = 3$) and basal cell-derived ALI samples (donor $n = 3$, control $n = 3$) and subsequently tested the effect of BMP inhibition. In contrast to the scattered distribution along normal tracheal epithelia, BMP2-expressing cells were significantly increased in TO biopsies, particularly concentrated in metaplasia regions and accounted for over 50% of all epithelial (E-Cadherin+) cells (Fig. 4a, Supplementary Fig. 6a–c). A similar result was recapitulated by TBBC-derived ALIs (Fig. 4b), showing an increased number of BMP2 positive cells in TO structures compared to non-TO controls. In regard to BMP signaling, phospho-Smad 1/5 was rarely seen during tissue homeostasis but evidently demonstrated in TO biopsies detectable in approximately 57% epithelial cells mainly at basal and intermediate layers and as well as stromal cells

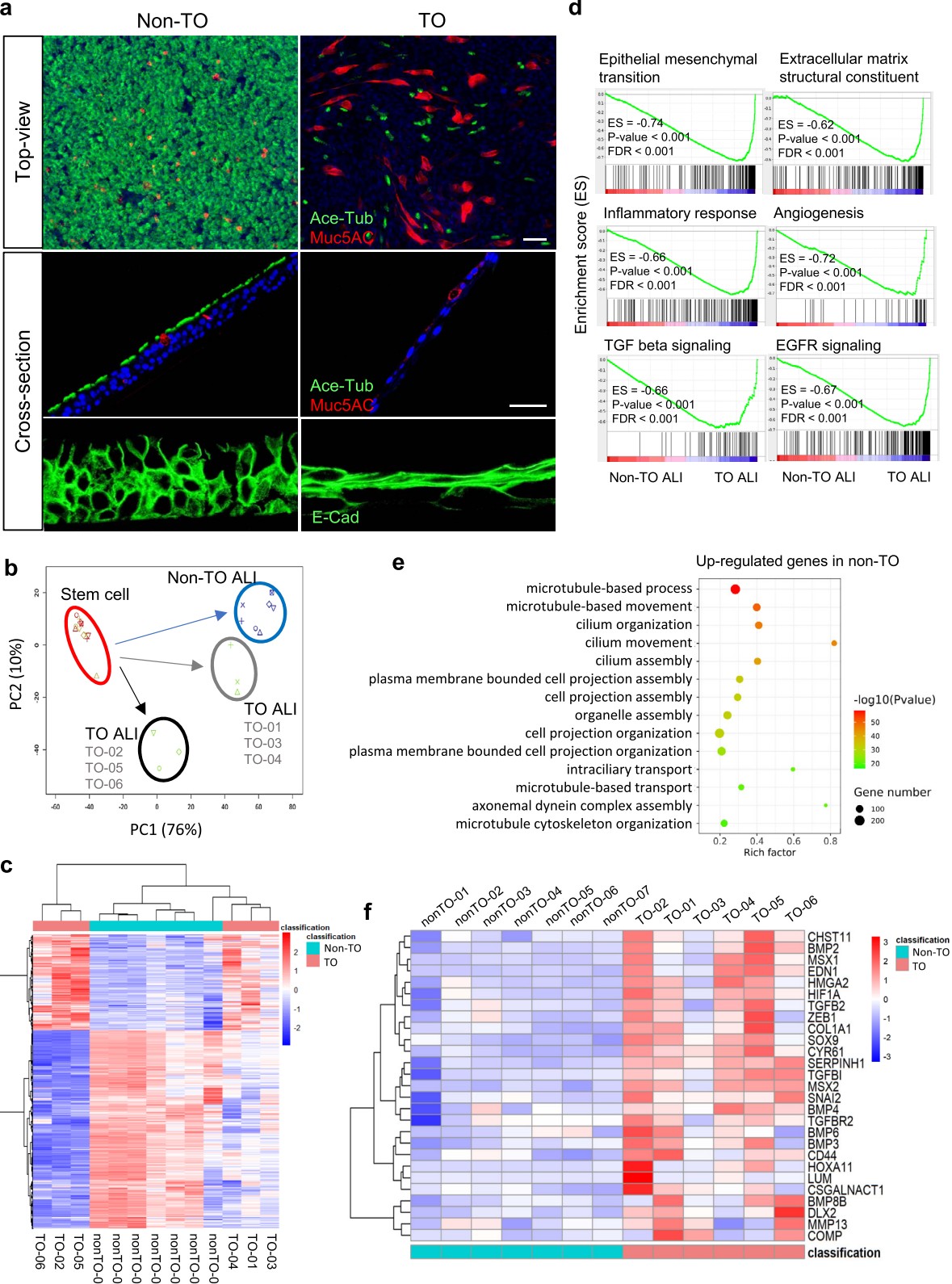

**Fig. 2 Differentiation potential of TO derived basal cells and transcriptional profiles of ALI structures. a** Basal cells underwent 20 days of differentiation on air–liquid interface (ALI) were stained with acetylated tubulin (Ace-Tub) and Muc5AC antibodies to reveal mucociliogenic efficiency (non-TO $n = 7$, TO $n = 6$ independent biological samples). E-Cadherin staining on cross-sections showing architectures of ALI structures. Scale bar, 50 μm. **b** Day20-ALIs were subjected to RNA-Seq. Whole-genome profiles of TBBCs before and after differentiation were displayed by PCA. **c** Differential expression between non-TO control-ALIs and TO-ALIs was visualized by heatmap with hierarchical clustering. **d**, **e** Gene Set Enrichment and GO enrichment analyses of RNA-Seq datasets derived from control and TO-ALIs. **f** Heatmap of genes involved in the axis of cartilage development that differentially expressed between control and TO groups.

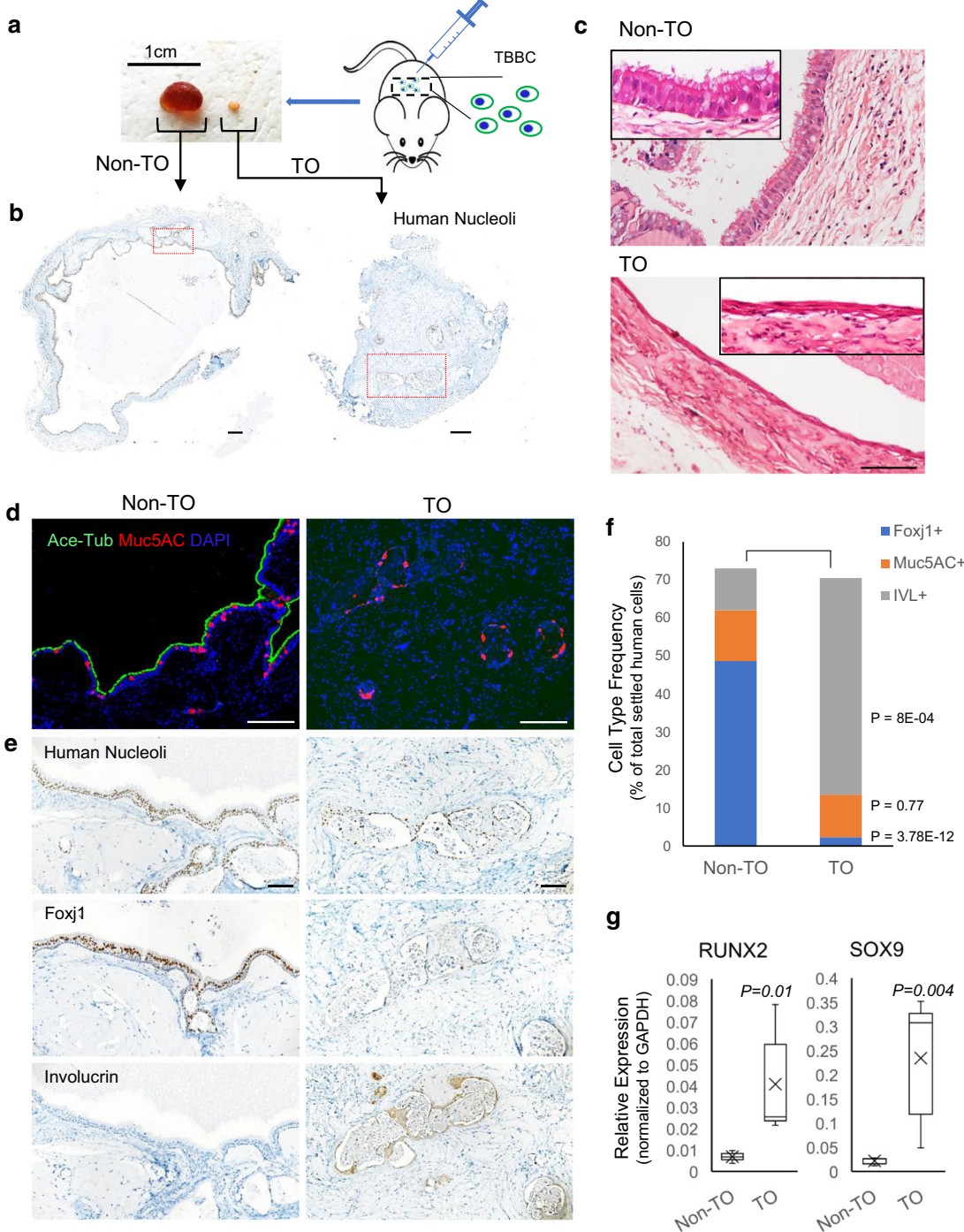

**Fig. 3 In vivo differentiation of TO-derived basal cells. a** Xenografts were standardly collected at 4-week post subcutaneous injection of $3 \times 10^6$ cells/spot on immunodeficient NSG mice. **b** Transplanted cells were distinguished from the host by immunohistochemical staining with anti-human nuclei. Scale bar, 250 μm. The red outlined boxes are the magnified regions shown in (**d**–**e**). **c** Hematoxylin-eosin (HE) staining of xenografts derived from control- and TO-basal cells. Scale bar, 100 μm. **d**, **e** IF and immunohistochemical (IHC) staining of functional cell markers Ace-Tub (ciliated), Muc5AC (goblet), Foxj1 (ciliated precursor), and Involucrin (squamous) on sequential xenograft sections. Scale bar, 100 μm. **f** Quantification of differentiation frequencies of basal cells from control and TO groups (non-TO $n = 7$ and TO $n = 10$ biologically independent experiments). Results are represented as mean ± SD, and p values are indicated to show statistical significance based on two-tailed Student's t-tests. **g** Real-time PCR measurement of RUNX2 and SOX9 expression on control- and TO-basal cell-derived xenografts ($n = 3$ biologically independent samples/group, 2 experimental repeats/sample). Boxes denote 25th and 75th percentile, lines denote median, crosses denote mean values, and whiskers min–max. P values are indicated to show statistical significance. Source data are provided as a Source Data file.

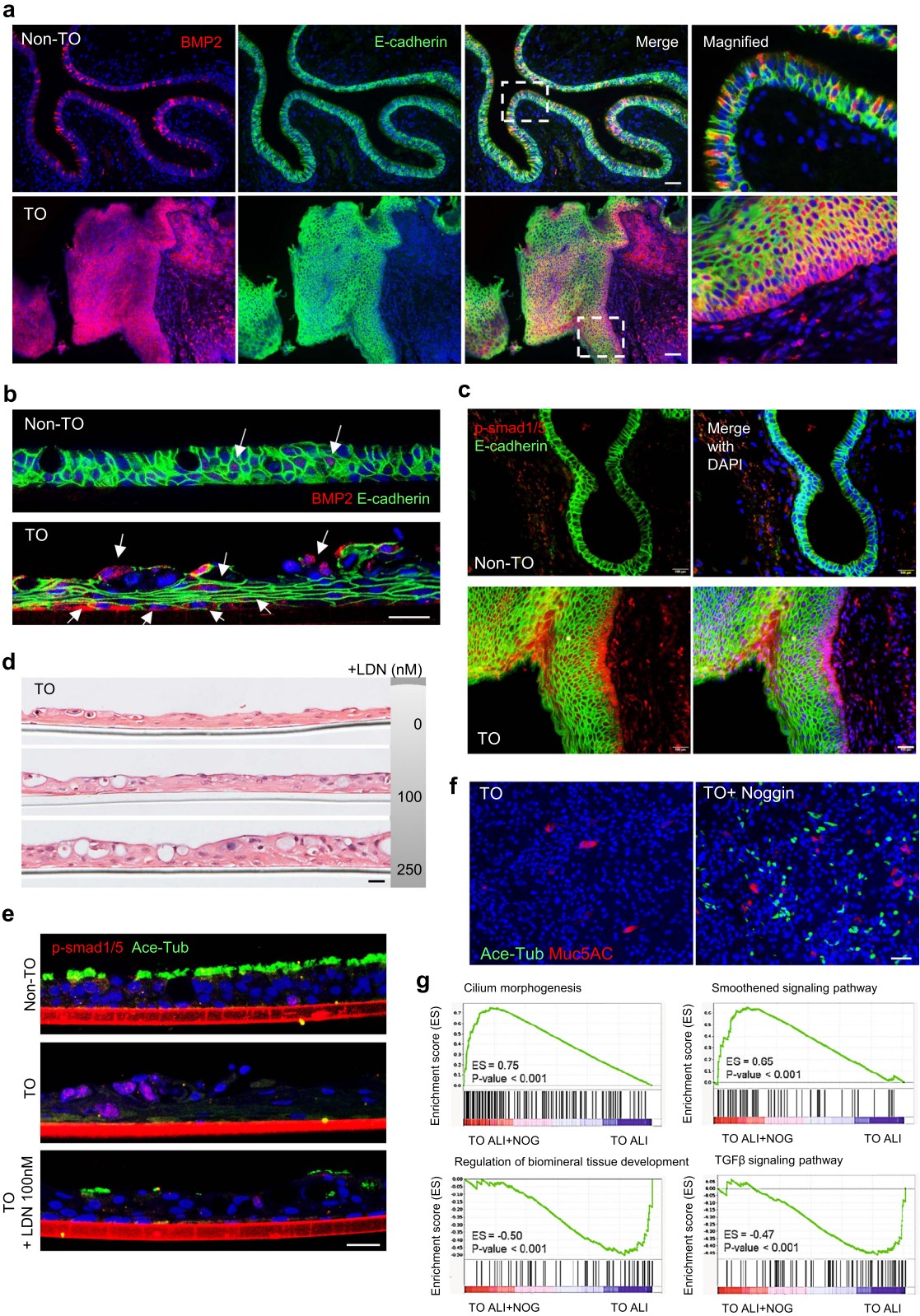

underneath (Fig. 4c, Supplementary Fig. 7a). On these biopsies, augmented TGFβ/phospho-Smad 2 signalings were also found in the TO cases, detectable in both epithelial (approximately 50% of total E-Cadherin+ cells) and stromal compartments (Supplementary Fig. 7a, b).

Treatment of TBBCs with LDN193189, a selective BMP receptor inhibitor, since Day 0 of differentiation resulted in a dose-dependent tissue recovery effect, enabling a switch from squamous (stratified)-like back to a pseudostratified structure (Fig. 4d, Supplementary Fig. 8a). In detail, ciliogenesis emerged at

**Fig. 4 BMP2 expression and activation of TGFβ-BMP signaling on TO-derived epithelia. a** IF microscopy of BMP2 staining on bronchoscopic biopsies collected from non-TO control and TO cases. Epithelial compartments were visualized by E-Cadherin co-staining. Scale bar, 100 µm. **b** Representative IF imaging of BMP2 on TBBC-derived ALIs, with E-Cadherin outlining the structures. BMP2 expressing cells were highlighted by white arrows. Scale bar, 20 µm. **c** Detection of phosphorylated SMAD 1/5 via IF staining on bronchoscopic biopsies collected from control and TO cases. Epithelial compartments were visualized by E-Cadherin co-staining. Scale bar, 50 µm. **a–c** *n* = 3 biologically independent experiments. **d** TO-TBBCs were treated with LDN-193189, a selective BMP signaling inhibitor, at a range of concentrations as indicated during ALI differentiation, and structures of cross-sections were visualized by HE staining (*n* = 3 independent experiments). Scale bar, 20 µm. **e** Ace-Tub and p-smad1/5 were assessed on ALIs with or without LDN-193189 treatments via IF staining (*n* = 1 dataset to verify LDN inhibition efficacy). Scale bar, 20 µm. **f** TO-derived pedigrees were treated with Noggin (200 ng/ml) constantly for 13 days throughout the course of ALI (*n* = 2 pedigrees). Mucociliary differentiation efficiency was assessed via IF staining and representative top-view images were shown. Scale bar, 50 µm. **g** Gene expression profiles of ALIs with or without Noggin treatment were examined by RNA-Seq, results of gene set enrichment analysis were shown as indicated.

a treatment concentration of 100 nM and reached a plateau since 250 nM (Fig. 4e, Supplementary Fig. 8b). When compared to non-TO normal controls, higher percentages of goblet cell differentiation often appeared in TO groups after BMP inhibition. This is explainable by the aforementioned inflammatory cell characteristics associated with TO-derived basal cells and resulting ALIs. Ciliary recovery could also be observed on TO ALIs when treated with 200 ng/ml Noggin, a protein antagonist blocking the binding of BMPs to receptors (Supplementary Fig. 9a). To exclude the possibility that the resultant restoration is merely due to a biased expansion of basal cell subpopulations rather than a change in cell property, we proceeded with Noggin treatment on TO pedigrees, which were established by single-cell cloning and from the same donors tested in LDN193189 treatment. As shown in Fig. 4f, emerged cilia were detected on 13-day ALI in the presence of Noggin. At a transcriptional level, there was an obvious shift in gene profile detected upon BMP inhibition (Fig. 4g), showing upregulated genes mainly related to cilium morphogenesis and smoothened signaling (a pathway important for airway epithelial development and executing synergistic effect on BMP suppression[37]), whilst a down-turn of genes related to cartilage-bone morphogenesis and unsurprisingly TGFβ signaling. Another observation that attracted our attention was that pre-treated TBBCs (with Noggin during cell expansion) were unable to recover mucociliary differentiation unless continuous inhibition was applied on ALI assay (Supplementary Fig. 9b). All these provide evidence for the existence of a pathogenic memory carried by TO-TBBCs to self-promote BMP production and TGFβ/BMP/SMAD signaling during basal cell differentiation. Based on current knowledge, epithelial metaplasia that is frequently found in patients with TO is actually an indication of abnormal BMP activation in the local microenvironment.

**Active effect of TO-derived epithelia on chondrogenesis.** In order to directly demonstrate the potential of TO-TBBCs in triggering osteo-chondrogenesis, we established in vitro (Fig. 5a) and in vivo (Fig. 5d, e) co-culture assays to allow close interactions between TBBC-derived epithelial and mesenchymal cells that mimics a physiological environment. By in vitro approach, wild-type human bone marrow-derived multipotent stromal cells (MSC) were pre-induced in chondrogenic media and then maintained in a basic medium without chondrogenic induction throughout the course of the assay in the presence of either non-TO or TO-derived cells on ALI. It has been demonstrated that MSC in TO group expressed higher levels of *SOX9* and *ACAN* compared to those in the non-TO group, and kept an upward trend through the course of co-culture (Fig. 5b). This was further evidenced by morphological assessment via Hematoxylin–Eosin (HE) staining and alcian blue-fast red staining. In comparison to MSC maintained in basic medium without TBBC co-culture and MSC in the nonTO-TBBC group, MSC when co-cultured with TO cells was more prone to form cartilaginous spheroids,

reflecting on morphological change from a solid compact aggregate into a differentiating spheroid with increased intercellular space and positive staining with Alcian blue (Fig. 5c, Supplementary Fig. 10a). A positive control under standard full induction was included in the assay to ensure MSC capability. On the other side, osteogenic lineage characterized by *RUNX2* and *BGLAP* induction was demonstrated in neither group (Fig. 5b), and no sign of extracellular calcium deposits was detected (Supplementary Fig. 10b). Taken together, these observations suggest that TO-TBBCs and their derived epithelium are beneficial for mesenchymal chondrogenesis but less likely ossification, at least under assay conditions.

Through in vivo approaches, we demonstrated that transplanted MSCs in a form as shown in Fig. 5d tended to aggregate around the epithelial nests derived from transplanted TO TBBCs and could be easily identified in a ring structure distinguishable from surrounding stroma by either HE or Alcian blue staining (Fig. 5f). In contrast, MSC aggregation was barely detected in the case of co-transplantation with non-TO cells, and only epithelial secretory cells have been recognized by Alcian blue staining due to mucin expression. FN1 is intensively expressed during MSC aggregation, condensation, and determination in the process of chondrogenesis, acting as a scaffold for MSC adhesion and differentiation[30]). When MSC is co-transplanted with TO cells, this protein has been found to accumulate the most in close proximity to the epithelial nests, then spread out in a radial pattern, which matches well with the distribution of transplanted MSCs (Supplementary Fig. 11a, b). In comparison, no such FN1 reactivity was detected under the MSC-nonTO-TBBC condition. As a potent chondrogenic growth factor, intense BMP2 expression was detected in both transplanted epithelial and surrounding MSC-ring structures (Supplementary Fig. 11c). From here, we demonstrated an active role of TO-TBBCs in ectopic chondrogenesis.

In order to further validate the effect of TO-TBBCs in chondrocyte differentiation, another co-transplantation approach, by subcutaneously inoculating pre-aggregated MSCs instead of cell suspension, was performed as shown in Fig. 5e. Animals were sacrificed at week 4 post-co-transplantation and examined for the presence of ectopic cartilage. As a result, morphologically mature cartilaginous spheroids were detectable under MSC-TO-TBBC condition, which were later confirmed by Alcian blue staining showing bright positive in target regions (Fig. 5g). Meanwhile, there was no cartilaginous growth found in circumstances either transplanted with aggregated MSC alone or co-transplanted with nonTO-TBBCs (Supplementary Fig. 12a). Based on all the above, we next compared the nodules acquired from non-TO- and TO-TBBC transplantations (Supplementary Fig. 12a, b). It has been found that parathyroid hormone-like hormone (PTHLH), which showed higher transcriptional levels in TO basal cells (2.78-fold↑ versus non-TO, *p*-value = 0.001) and TO ALIs (10.53-fold↑ versus non-TO, *p*-value = 1.24E−08) in forementioned RNA-Seq data, was evidently increased in TO-nodules (Supplementary Fig. 12c).

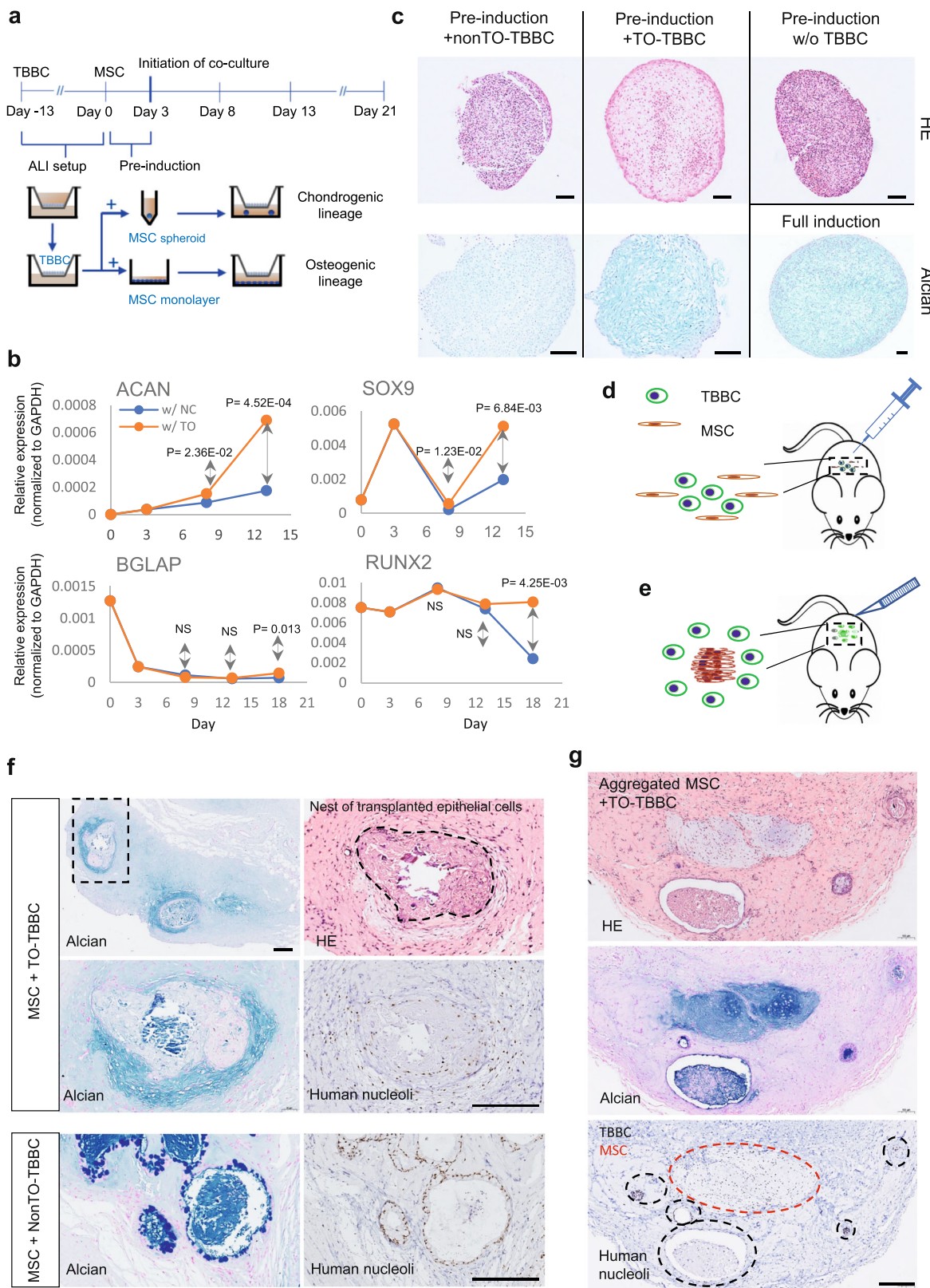

They expressed in both epithelial and a large number of stromal cells, whilst in non-TO control, this protein was barely detectable in epithelial and occasionally seen in the stroma. With previous knowledge about PTHLH as a well-documented regulator of osteo-chondrogenesis[38], its elevation in TO nodules enhances current findings and further demonstrates an active role of TO-TBBC in chondrogenesis by stimulating mesenchymal cells and modulating the local microenvironment.

**Fig. 5 Co-culture assay of TBBC and human mesenchymal stem cells (MSC). a** Flow chart illustrating in vitro assay setup. TBBCs were pre-differentiated for 13 days on air-liquid interface prior to co-culture, on the other side, MSCs were pre-induced as pellets in chondrogenic medium or pre-induced as monolayers in osteogenic medium for 3 days before co-culture with TBBCs. RNA of MSC samples were collected every 5 days post-co-culture for PCR and fixed at the endpoint (Day 21) for staining. **b** Real-time PCR measurement of SOX9/ACAN (Chondrogenic-related) and RUNX2/BGLAP (Osteogenic-related) gene expressions on MSCs at D5, D10, and D15 post co-culture ($n = 2$ experimental replicates per group). w/nc denoting co-culture with non-TO control, w/TO denoting co-culture with TO-TBBC. Two-tailed Student $t$-tests were performed and $p$ values are indicated to show statistical significance. **c** HE and Alcian Blue staining on MSC spheroids which were co-cultured with control or TO cells for 21 days, to reveal the progress of chondrogenesis ($n = 2$ independent experiments). MSC spheroids cultured alone under basic conditions and spheroids under standard differentiation conditions were examined in parallel as negative and positive controls. Scale bar, 100 µm. **d, e** Schematic diagrams showing two in vivo approaches to perform TBBC-MSC co-culture assay. Dual-cell transplantation onto immunodeficient NSG mice was performed either by subcutaneous injection of TBBCs ($2 \times 10^6$ cells/spot): MSC ($2 \times 10^6$ cells/spot) mixture resuspended with 50% Growth factor-reduced Matrigel in a total volume of 200 µl/spot, or via minor surgery with aggregated MSC ($10^6$ cell/spheroid) surrounded by a gel-like layer of TBBCs resuspended in Matrigel. Cartoon mice in this graph were adapted from the online source clipartcraft.com. **f** HE and Alcian Blue staining on xenografts collected from approach D ($n = 2$ independent experiments). The human origin of growths found in nodules was confirmed by IHC staining with an anti-human nucleoli antibody. Scale bar, 200 µm. **g** HE and Alcian Blue staining on xenografts collected from approach E ($n = 2$ independent experiments). The human origin of growths was confirmed by IHC anti-human nucleoli staining. Scale bar, 200 µm. Source data are provided as a Source Data file.

---

**Evaluation of TBBCs from nodule-free regions in TO patients.** To better understand the incidence and distribution of pathogenic basal cells under TO conditions, we extended the study on TBBCs that were isolated from relatively normal (determined under bronchoscopy) tracheobronchial epithelial in TO patients. Unlike monodirectional changes found in lesion derived-TBBCs among donors, basal cell condition varied from case to case in patient-matched (PM) normal regions, presented in three distinct phenotypes by in vivo differentiation assay (Fig. 6a, Supplementary Figs. 13–15). Being in line with their ALI expression profiles, TBBCs were spontaneously clustered into three groups depending on TO stage (Fig. 6b). Specifically, basal cells from stage I donors were shown to undergo normal-like (NL) differentiation and express club cell- and ciliated cell-related genes, whilst those from stage II donors were prone to goblet cell metaplasia (GM) with gene expression related to both ciliogenesis and immune response when proceeded to stage III, PM-TBBCs appeared similar to their counterpart TO-TBBCs showing cilium dysplasia (CD) and with significant enrichment of genes involved in processes of inflammation, extracellular matrix organization, angiogenesis, and skeletal development (Fig. 6c). These results, together with aforementioned observation of increased goblet cell differentiation in BMP inhibition treatment (Supplementary Fig. 8b), imply that prolonged inflammatory signaling (or called "inflammation memory") in TBBC is likely to be a grounding change in TO. This happened at early stages in epithelial regions where nodules have not yet developed.

**Correlation of TBBC malfunctions with epigenetic changes.** Thus far, our data suggest the presence of sustained changes in TO-derived TBBCs. We further wondered whether these changes in gene expression signature were owing to a pre-set landscape of chromatin accessibility. ATAC-sequencing was performed to measure the accessible genome in TBBCs from PM and lesion regions of two TO donors (TO-01, stage I; TO-02, stage III) and two non-TO controls. The Pearson correlation analysis indicated that cells from non-TO cases and PM region of Stage I case were sharing the strongest correlation, while cells from lesion-region of two TO cases and PM region of Stage III case were separately clustered (Supplementary Fig. 16a). This enhances our finding from studies of PM-TBBCs. However, we noted that inter-cluster variations were not statistically larger than intra-cluster variations for TO cases. One possibility for this is that TO-associated changes may confine at a limited number of loci, therefore there are still high similarities between disease and normal cells. To better understand the extent to which epigenetic changes may

contribute to disease phenotype, six samples were divided into two main groups (non-TO versus TO) based on their differentiation phenotype and clustering results. Non-TO group shares similar degrees of whole-genome accessibility, indicated by total peak number, but which is more diverse in TO (Fig. 7a). Similar read distributions with enrichment at the proximity of the transcription start site (TSS) were confirmed in both groups, indicating a successful generation of open region-specific data for the following analysis (Fig. 7b). Motif enrichment analysis of differential peaks (i.e. 2,22,914 up and 6406 down in TO versus non-TO) was performed to overview the potential difference in the regulatory network between samples. As shown in Table 2, TO-derived TBBCs and Non-TO are under the control of distinct panels of TF. Particularly, a downward trend of p63, p73, Six, and Fox (respiratory specialization-related TFs) binding whilst increased putative binding with Ets1, SPDEF, Runx, and Jun-B (TFs involved in mesenchymal differentiation, mucosal homeostasis, skeletal and skin development) was revealed in TO (Supplementary Fig. 16b).

To functionally interpret the variations in chromatin accessibility and their association with gene regulation, differential peaks were annotated by the nearest genes (Fig. 7c), then were proceeded with joint-analysis with ALI RNA-Seq of corresponding samples followed by pathway enrichment (Fig. 7d). The joint analysis of basal cell ATAC-Seq and differentiated ALI RNA-Seq was aiming to identify pre-existing chromatin change that influences gene expression in the process of stem cell differentiation. Of 1801 gene loci associated with increased accessibility in TO (FC > 2 versus non-TO), 908 (50%) loci displayed a positive correlation with their gene expression (RNA FC > 2 versus ALI non-TO) (Quadrant 1 in Fig. 7d), referring to genes involved in inflammation-related and mucin production pathways. There were 893 gene loci associated with less accessibility in TO, of which 569 (64%) loci displayed a positive correlation with gene expression and enriched in pathways related to club and ciliated cell differentiation and metabolism (Quadrant 3 in Fig. 7d). Zooming into chromatin changes based on altered domain (Supplementary Table 2-6), it has been noted that key players of ciliogenesis such as *FOXJ1*, *TFF3*, *TEKT2*, *CBY1*, and *RSPH9* were intrinsically less accessible in TO-TBBCs at either their promoter-proximal (<1 kb from TSS) or first intron regions (Fig. 7e, Supplementary Fig. 17a), which provides a possible explanation for the transcriptional suppression on these genes during subsequent differentiation (Supplementary Fig. 17b). On the other hand, a pre-active status was detected in TO basal cells on a number of genes essential for immune response (such as

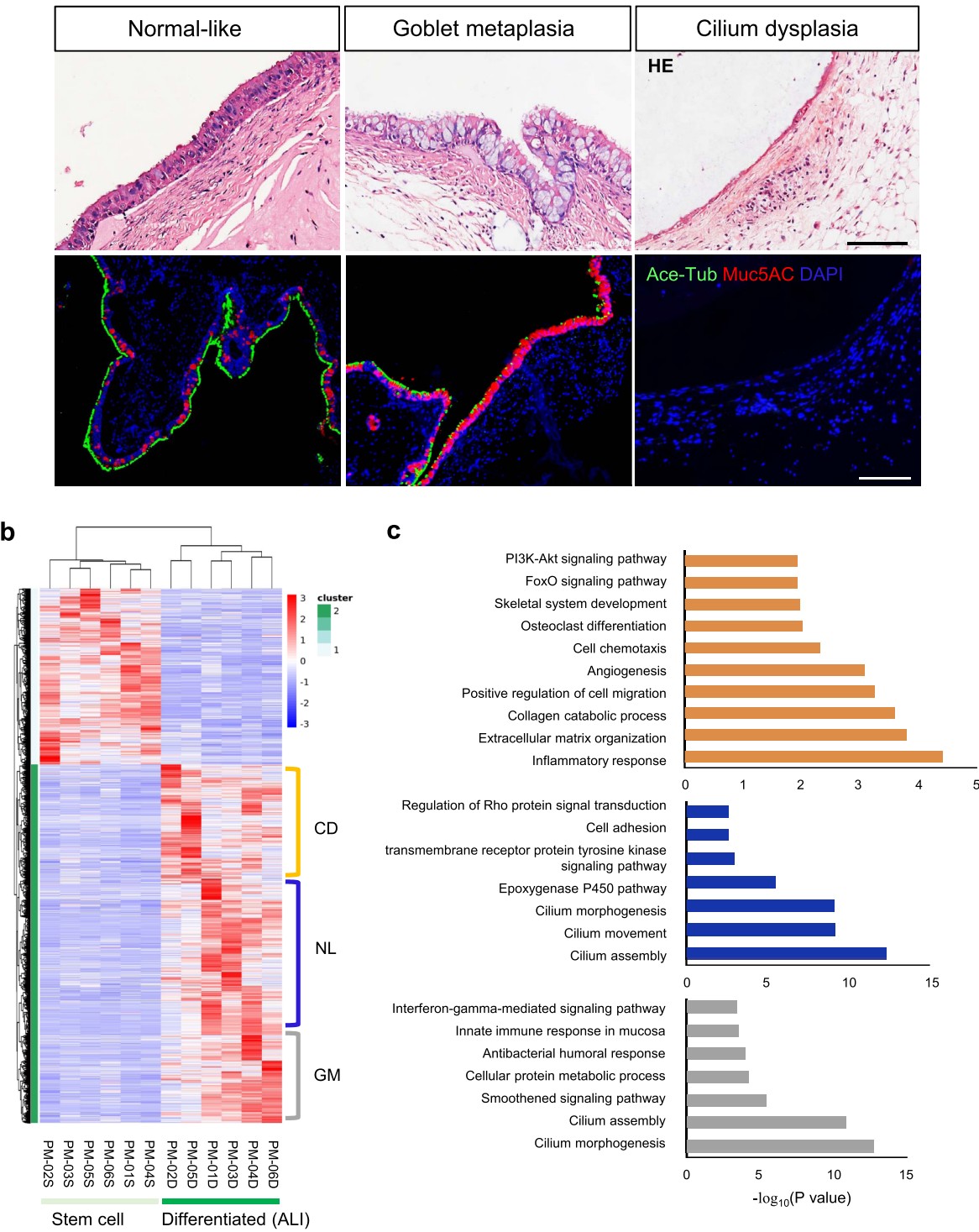

**Fig. 6 Characterization of basal cells derived from normal-looking regions of TO patients. a** HE staining showing distinct cell commitments of patient-matched (PM) normal-looking region-derived TBBCs (PM $n = 6$ independent biological samples). Differentiation efficiency was assessed by IF staining on xenograft sections with Ace-Tub (ciliated) and Muc5AC (goblet) cell-type-specific markers. Scale bar, 100 μm. **b** PM-TBBCs prior to and post 20-day ALI differentiation were subjected to RNA-Seq. Differential gene expression profiles were visualized by heatmap with hierarchical clustering. **c** GO and KEGG enrichment analyses were performed to reveal molecular characteristics of three sub-groups, corresponding to normal-like (NL), goblet-metaplasia (GM), and cilium dysplasia (CD) phenotypes.

IDO1[39,40]), extracellular matrix metabolism, and skeletal development (such as ODAPH, CSGALNACT1, and BMP3[41–43]), corresponding to the previous observation of highly induced expression on these genes during differentiation but which was subtle in normal controls (Supplementary Fig. 17a, b). All these suggest there are pre-existing chromatin alterations in TBBCs, upon their receiving cues from the local milieu under TO conditions.

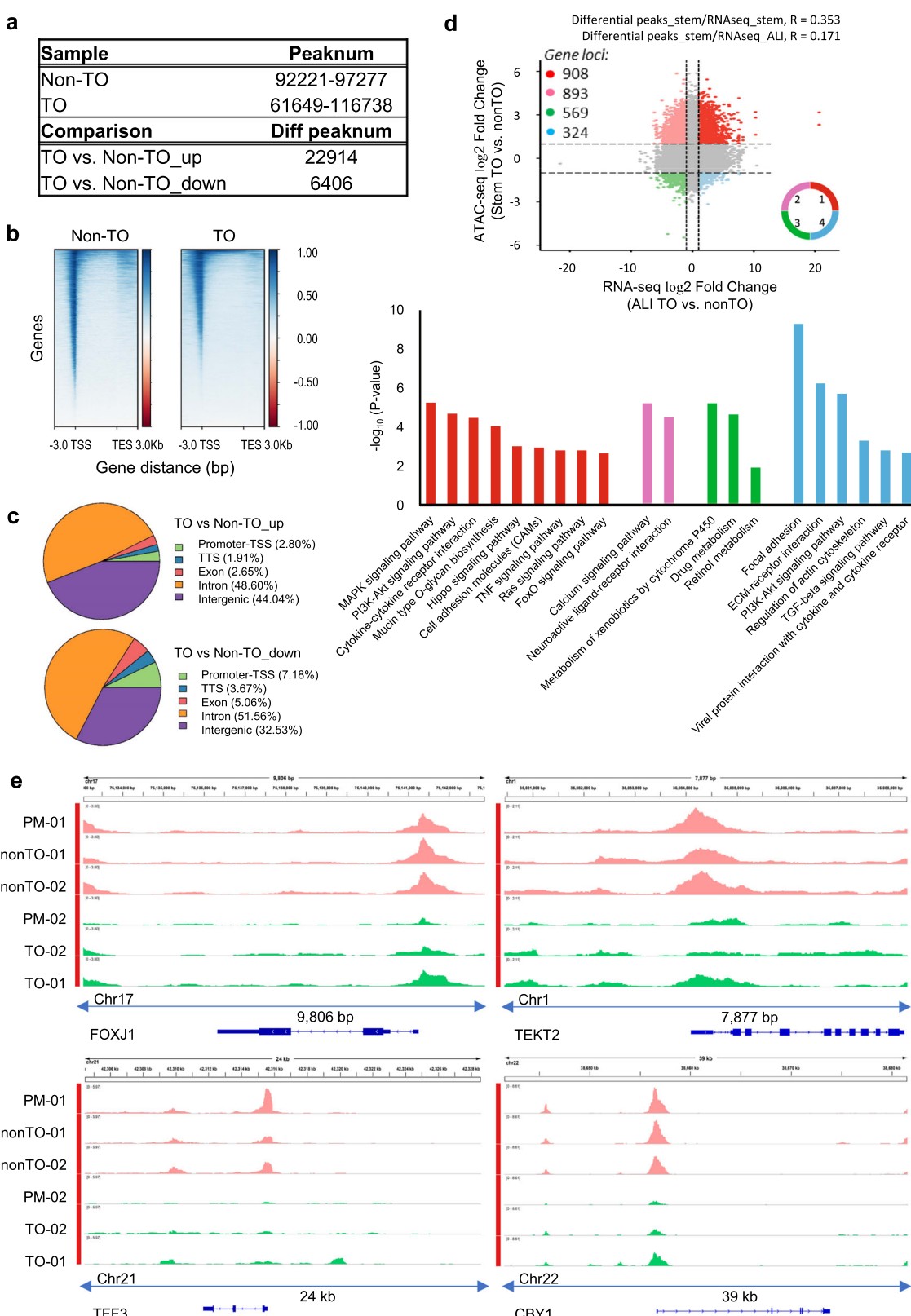

**Fig. 7 Chromatin accessibility profiles of TO-derived basal cells. a** Total peak number and differential peak number in TO versus control. **b** Heatmap showing reads distributions (from bigwig) across the gene. Genes are represented as lines that are sorted in a descending order based on signal intensity. TSS transcriptional stat site, TES transcriptional end site. **c** Differentially expressing peaks were annotated and show the distribution in pie charts. **d** Four-quadrant graph showing the overlaps of genes with altered accessibility and corresponding changes on expression. Vertical and horizontal dotted lines: log2FC = ±1. Pearson's *R* was calculated between two groups, stem cell ATAC-Seq against stem cell RNA-Seq and stem cell ATAC-Seq against differentiated (ALI) cell RNA-Seq. Pathway enrichment analysis was performed on genes from Quadrant 1–4. **e** IGV plots depicting ATAC-Seq signals at selected gene loci. Peak height is positively correlated to accessibility. Genome-scale is included and the *y*-axis is consistent for all loci.

**Table 2 Top 20 most enriched TF motifs in Non-TO and TO-derived basal cells.**

| Non-TO | | | TO | | |
|---|---|---|---|---|---|
| Transcription factor | % of Targets with motif | P-value | Transcription factor | % of Targets with motif | P-value |
| p63 | 15.75% | 1e−383 | FRA1 | 15.85% | 1e−798 |
| p53 | 6.87% | 1E−258 | BATF | 17.67% | 1e−756 |
| p73 | 4.50% | 1E−227 | ATF3 | 17.93% | 1e−744 |
| FRA1 | 10.97% | 1E−126 | JUNB | 15.66% | 1e−730 |
| FRA2 | 9.71% | 1E−123 | FRA2 | 13.76% | 1e−722 |
| FOSL2 | 7.43% | 1E−119 | AP-1 | 18.80% | 1e−683 |
| ATF3 | 12.11% | 1E−115 | FOSL2 | 9.92% | 1e−590 |
| JUNB | 10.55% | 1E−115 | JUN-AP1 | 7.55% | 1e−501 |
| BATF | 11.75% | 1E−108 | EHF | 19.85% | 1e−201 |
| AP-1 | 12.72% | 1E−102 | ELF3 | 12.94% | 1e−189 |
| JUN-AP1 | 5.62% | 1E−101 | ELF5 | 12.43% | 1e−181 |
| NF1-halfsite | 27.01% | 1E−87 | ELF4 | 14.27% | 1e−148 |
| SIX1 | 5.21% | 1E−62 | ETS1 | 12.89% | 1e−102 |
| SIX2 | 14.66% | 1E−60 | GRHL2 | 6.40% | 1e−99 |
| NF1 | 7.87% | 1E−53 | ERG | 20.67% | 1e−97 |
| FOX | 14.47% | 1E−47 | ETV1 | 16.40% | 1e−91 |
| FOXM1 | 16.36% | 1E−46 | SPDEF | 12.70% | 1e−81 |
| FOXA1 | 34.23% | 1E−44 | ETV2 | 12.18% | 1e−69 |
| TEAD4 | 13.10% | 1E−43 | GABPA | 10.48% | 1e−68 |
| TEAD1 | 14.77% | 1E−41 | RUNX | 9.02% | 1e−58 |

Motif enrichment analysis of differential ATAC-Seq peaks in TO versus Non-TO by HOMER. *P* values were calculated following findMotifsGenome.pl algorithm.

## Discussion

Erosion of respiratory mucociliary epithelium, accompanied by squamous metaplasia, abnormal submucosal chondrification, and ossification, as well as inflammatory infiltration are key histologic features of TO[16]. Central questions of how these cartilaginous/osseous nodules develop, and the extent to which epithelial alterations correlate with pathological changes in the subepithelial stroma zone are still far from settled. Our present work, from a view of epithelial stem cells- a foundation element of tissue structural and functional maintenance, provides an insight into the mechanism underlying TO pathogenesis.

We demonstrate herein, an appearance of malfunctioning basal cell populations under TO conditions, which is not only confined at lesion regions but also showing a more prevalent existence in nearby normal-like epithelium. There are two types of abnormalities detected on patient-derived basal cells, referring to a goblet-cell-hyperplasia phenotype in a population derived from lesion-free regions of mid-stage TO and a more prominent ciliary dysplasia, squamous-prone phenotype displayed by basal cells derived from lesion regions and as well as lesion-free regions of Stage III TO. Early appearance of goblet cell-biased fate determination in lesion-free region implies a grounding change on tracheobronchial basal cells prior to nodule development. Inflammatory characteristics have been evidently detected in TO-TBBCs, reflecting on cytokine-chemokine expression signatures and pre-active chromatin landscapes enriched for AP-1 (an immune activator) and SPDEF (a master regulator of goblet cell differentiation) binding motifs. This supports the theory of involvement of chronic inflammation in TO exacerbation and explains well the symptomatic exhibition of sputum production often found in these patients.

Sustained activation of the TGFβ-BMP axis is considered as a second change that occurred in TBBCs during TO progression. Although NRG1/PI3K and AREG/EFGR signaling (known contributors in goblet hyperplasia)[34,44] were shown to be upregulated in TBBCs derived from nodule-containing regions, these cells did not exhibit secretory cell-prone differentiation unless a BMP inhibition treatment was applied. This observation, being in line with previous findings that the TGFβ-BMP-SMAD axis exerted an inductive effect on squamous metaplasia and contrarily restricted mucociliary differentiation[45,46], indicates a sequential event of basal cell alteration when tissue develops into a nodule formation stage. The ciliary dysplasia squamous-prone phenotype demonstrated in TO-TBBCs is attributed to signal superimposition.

Upon two-folded changes, more striking alterations were observed on malfunctioning TBBCs. On top of the aforementioned change in terms of epithelial character, these cells exhibited a strong potential in epithelial-to-mesenchymal transition and act as a putative matrix modulator and a local supplier of BMPs and TGFβ-related family members. After in vitro differentiation, a wide panel of mesenchymal cell hallmarks, such as N-cadherin, Vimentin, Slug, ETS-1, and α5β1 integrin as well as fibroblast activation protein, were significantly induced in TO expression profile, and accompanied by the production of a wide range of collagens and activation of a list of BMP target genes encoding for chondro-osteoinductive transcription factors, such as MSX2, SOX9, RUNX2, and distal-less homeobox proteins DLX family. Nevertheless, there is no direct evidence showing a terminal chondrocyte-osteoblast differentiation of TO-derived basal cells, demonstrated by a lack of Type II collagen, ACAN (aggrecan), ALPL (biomineralization associated alkaline phosphatase), and BGLAP (osteocalcin) expression, although a report of osteogenic differentiation of human lens epithelial cells was previously shown[47]. These results lean towards a viewpoint that TO-basal cells are likely to serve as a trigger for submucosal chondro-osteogenesis through a secretion manner, by maintaining high concentrations of local BMP/TGFβ ligands and orienting stroma environment towards a skeletal trophic status, for instance, production of odontogenesis associated phosphoprotein, chondroitin sulfate N-acetylgalactosaminyltransferase, parathyroid hormone-like hormone and fibronectin that play indispensable functions in cartilage and bone formation[38,42,43,48].

Based on the current knowledge about spatial and temporal regulation of BMP signaling and its cooperation with the TGFβ pathway during skeletal development and homeostasis[41,49,50], a threshold of BMP3 is required for the onset of chondrogenesis and excess BMPs has been reported to associate with expanded skeletal elements and joint fusion via enhancing chondrocyte

proliferation but delaying osteoblast maturation[51]. Meanwhile, a constant high dose of environmental TGFβ through the course of chondro-osteogenesis would also result in abnormal skeletal mass via increasing the pool of pre-osteoblast progenitors and reducing osteoclast differentiation. Taken together with the nature of differentiated TO-TBBCs as a repository for BMPs and TGFβ signals, it may provide an explanation for the detection of chondrogenic but less likely osteogenic differentiation of MSC when co-cultured with epithelia derived from TO-TBBCs. To work in concert with the existing theories of TO etiology referring to ecchondrosis/exostosis from tracheal rings and metaplasia of connective tissue[52–55], a cooperative relationship between pathogenic basal cell and TO-derived stroma is worthy of further investigation.

On top of well-documented TGFβ-BMP functions in osteo-chondrogenesis, proinflammatory cytokines have also been reported to exert important roles in skeletal homeostasis and bone remodeling[56–58]. IL-1, TNFα, and IFNγ are recognized contributors to BMP2 induction in cell models[56,57,59,60] and the inflammation itself is able to induce TGFβ-BMP activation in osteoclast cells as a compensatory response leading to osteoprogenitor cell recruitment and new bone formation under ankylosing spondylitis condition[57]. This is reminiscent of the coincidence of inflammation and TGFβ-BMP activation in TBBCs under TO. Taken together with current findings from gene profiles, chromatin accessibility, and phenotypic studies, early appearance of inflammatory memory in TO basal cells is speculated to act as an upstream event of latter TGFβ-BMP activation and this may explain why constant BMP inhibition, rather than pre-treatment or transient treatment with BMP inhibitors, was necessary for ALI recovery and there was still a trend of goblet hyperplasia upon BMP inhibition. Furthermore, a by-pass observation of accumulating TRAP (tartrate-resistant acid phosphatase[61])-positive cells at the xeno-sites in TO TBBC-MSC co-transplantation animal models, further evidence a role of TO basal cells in evoking inflammation. The function of local inflammation in MSC activation may enhance basal cell-mediated effects on ectopic osteochondrogenesis and needs to be further elucidated in TO.

In conclusion, our study expands the current knowledge about TO pathogenesis and reports a type of stem cell malfunction in the aspect of promoting neoosteogenesis. These findings may shed a light on further understanding of various pathological conditions, such as neoosteogenesis in chronic rhinosinusitis and pulmonary ossification in IPF[62,63].

## Methods

**Patient population and sample collection.** Patients with an established diagnosis of TO were confirmed by CT scan and bronchoscopy with histopathologic findings. Non-TO controls were those subjects undergoing surgery for lung cancer or bronchoscopy for interstitial lung disease (none of these controls presented nodular lesions, inflammation or infection of the trachea). Patients with TO were classified in terms of lesion type and stage according to characteristic bronchoscopic visualization as described previously[16,64]. Briefly, the disease could be categorized into three lesion types based on the extent of nodules in mucosa: scattered TO (few nodules with large areas of normal mucosa between them), diffuse TO (numerous nodules covering the entire mucosa with almost no normal area), and confluent TO (fusion of adjacent lesions); or divided into three stages based on nodule condition along lesion regions: Stage I (where scattered plaque-like yellow-whitish soft lesions are usually found), Stage II (where cobblestone- or stalactitic cave-nodules are projecting into the lumen) and Stage III (where the tracheal wall becomes deformed and rigid leading to airway narrowness even obstruction). The study protocol was approved by the Ethics Committee of Shanghai Sixth People's Hospital, and informed consent was obtained from all individual participants recruited for this study. The study design and conduct complied with all relevant regulations regarding the use of human study participants and was performed in accordance with the criteria set by the Declaration of Helsinki. Biopsy samples were collected by bronchoscopic brushing from diseased regions and lesion-free regions of patients with TO and trachea-bronchi from non-TO controls.

**Cell culture.** Brushing samples were washed in pre-chilled wash buffer (Ham's F12, 5% fetal bovine serum, 100 μg/ml pen strep, 100 μg/ml Gentamycin, and 0.25 μg/ml Fungizone), and digested with dissociation buffer (Ham's F12, 1 mg/ml collagenase IV (Gibco)) for 20–30 min at 37 °C with gentle rocking. Dissociated cells were washed thoroughly in cold wash buffer and plated on irradiated 3T3-J2 feeder layers in 12-well culture plate (2 wells/brushing), followed by 5–7 days culture in growth medium (3:1 DMEM/Ham's F-12, 10% fetal bovine serum, 5 μg/ml insulin (Sigma), 10 ng/ml epidermal growth factor (R&D Systems), 5 μM Y-27632 (Tocris), 100 ng/ml Noggin (R&D Systems), and 1 μM SB431542 (Tocris)) at 37 °C in a 7.5% $CO_2$ atmosphere for each passage.

Clonogenic epithelial cells from cultures were purified by negative selection using Feeder Removal Microbeads (Miltenyi Biotech). Clonogenicity assays were performed by seeding 2000 purified cells/cm²/passage on feeders and cultured for 4–5 days when used for counting. Colonies were manually quantitated under Nikon Eclipse Ts2 inverted microscope. Colony-forming efficiency was determined by counting the number of colonies with a size of ≥10 cells/clone in each culture and shown as a percentage of input epithelial cells.

Feeder-free monolayer culture of airway basal cells was performed by plating microbeads-purified clonogenic cells (passage 2–4) directly on the plastic surface at a density of 4000 cells/cm² under routine growth conditions as described above. Cell morphology was being monitored within a culture period of 5–7 days, then confluent cell layers were fixed for microscopy analysis.

**Differentiation assays.** For in vitro differentiation, microbeads-purified epithelial cells were harvested from cultures and plated on inserts of 6.5 mm transwell with 0.4 μm pore polyester membrane (Corning) at a density of $6 \times 10^5$ cells/cm², followed by 2–3 days expansion in growth medium then maintained as air–liquid interface in PneumaCult-ALI medium (StemCell Technologies) for 20 days until structures mature. Differentiated cells were fixed for IF staining and meanwhile, preserved in TRIzol for standard RNA isolation.

For in vivo differentiation, cultured cells ($3 \times 10^6$ cells/injection) were resuspended in Dulbecco's Modified Eagle's Medium and pre-mixed 1:1 (v/v) with growth factor reduced Matrigel (Corning) prior to subcutaneous injection on immunodeficient NOD-$Prkdc^{scid}Il2rg^{em1}$/Smoc mice (Shanghai Model Organisms). The 6–8-week-old male and female animals were used for all experiments. Mice were sacrificed at Week 4 post cell injection, and nodule growths were collected for paraffin embedding, sectioning and subsequently IHC/IF assessment. Meanwhile, differentiated structures were preserved in TRIzol for standard RNA isolation. All procedures were conducted under IACUC guidelines and approved protocol by the IACUC committee of Shanghai Sixth People's Hospital.

**BMP inhibition treatment.** Cultured cells were harvested and plated on inserts as described above for routine differentiation. BMP inhibitors Noggin (200 ng/ml) and LDN193189 (100, 250, and 500 nM) were added separately in PneumaCult-ALI medium since Day 0 of differentiation. Fresh ALI medium with or without inhibitor supplement was replaced every 2 days until structures mature. Upon harvest, growths on inserts were fixed for histological assessment and meanwhile, preserved in TRIzol for standard RNA isolation followed by expression analysis.

**Immunohistochemical and immunofluorescent analysis.** Cells grown on culture dish and differentiated structures were fixed in 4% (w/v) paraformaldehyde, proceeded to paraffin embedding for tissue sectioning or directly used for clone staining. Cells were permeabilized with 0.1% Triton X-100 in PBS containing 5% bovine serum albumin, and labeled with the primary antibodies as listed below: mouse anti-p63 (IF 1/100, ab735), mouse anti-Human nucleoli (IHC 1/200, ab190710), rabbit anti-Involucrin (IHC 1/500 dilution, HPA055211), rabbit anti-Ki67 (IF 1/1000, ab15580), rabbit anti-CK10 (IF 1/200, ab76318), rabbit anti-Muc5AC (IF 1/250, IHC 1/500, ab198294), rabbit anti-p75 NGF receptor (IF 1/50, ab52987), rabbit anti-BMP2 (IF 1/150, ab214821), mouse anti-Aggrecan (IHC 1/500, ab3778) and mouse anti-Osteocalcin (IHC 1/100, ab13418) purchased from Abcam, rabbit anti-CK14 (IF 1/400, 10143-1-AP) and rabbit anti-Fibronectin (IF 1/200, 15613-1-AP) acquired from Proteintech, rabbit anti-CK5 (IF 1/100, MA5-14473) and mouse anti-Uteroglobin (IF 1/200, MA5-17170) from Invitrogen, mouse anti-CK8 (IF 1/400, NBP1-48281) and rabbit anti-Foxj1 (IF 1/200, IHC 1/500, NBP1-87928) from Novus, rabbit anti-phospho-Smad1/5 (IF 1/500, #9516) and rabbit anti-phospho-Smad2 (IF 1/500, #18338) from Cell Signaling Technology, Goat anti-E-Cadherin (IF 1/200, AF648) from R&D Systems, rabbit anti-PTHLH (IF 1/150, DF6532) from Affinity and mouse anti-Acetylated Tubulin (IF 1/30,000, T7451) from Sigma. Appropriate Alexa Fluor 488 or 594 conjugated secondary antibodies (Abcam) were used for IF and Vector Labs ABC kit with DAB substrate (Vector Laboratories) was used for IHC. Cells were viewed and photographed using Leica DM6 B upright digital microscope (Leica Application Suite X software), Olympus IX53 (Olympus cellSens Imaging Software), and Zeiss Axio Vert.A1 inverted microscope (ZEN 2.6 lite).

For image analysis and quantification, immunohistochemical images were captured using NanoZoomer S210 Digital slide scanner, and the entire image was quantified by HALO v3.0.311.293 software (Indica Labs). Immunofluorescent images were captured as aforementioned, followed by FIJI quantification. Data are

presented as a percentage of total epithelial cells or a percentage of measured area depending on the distribution of target signals.

## Co-culture assay

*In vitro co-culture.* Human bone marrow-derived MSC was purchased from Fuyuan Biotechnology (Shanghai, China). Cells from P4 culture were used in experiments. To test for chondrogenic lineage, MSCs were harvested when reached 85% confluency, and resuspended in MSC Chondrogenic Differentiation medium (Fuyuan Biotechnology) then spun down into pellet ($1 \times 10^6$ cells/500 μl/tube) in 15 ml falcon tube. Cell pellets were kept at the bottom in falcon tubes without disturbance to allow the formation of spheroids (1 spheroid/tube). After 24-h, cell pellets were detached from the tube bottom and cultured in a fresh differentiation medium for another 48-h. MSC spheroids cultured for a total of 72 h were transferred to the outer well of the transwell and maintained in a basic medium (provided by the manufacturer) to initiate co-culture with Day13 pre-differentiated TBBC-ALIs. The basic medium was refreshed every 2 days during the course of co-culture. MSC spheroids continuously cultured in a differentiation medium without TBBC addition were examined as a positive control. To test for osteogenic lineage, MSCs were plated at $2 \times 10^4$ cells/cm$^2$ in the fibronectin-coated 6-well plate using MSC Osteogenic Differentiation medium (Fuyuan Biotechnology). After 72-h, Day 13 pre-differentiated TBBC-ALIs were translocated onto pre-set MSC culture wells to allow initiation of co-culture in basic medium (provided by the manufacturer). The medium was refreshed every 2 days as routine. MSCs at Day5, 10 post-co-culture along chondrogenic differentiation, and respectively at Day 5, 10, 15 post-co-culture along osteogenic differentiation were subjected to polymerase chain reaction (PCR) analysis. Endpoints of differentiated MSC at Day 21 post-induction were fixed for Alcian Blue and Alizarin Red S staining, respectively. Chondrogenic spheroids harvested at Day 21 proceeded with paraffin embedding and sectioning for further histological assessment.

*In vivo co-culture.* For the suspension approach, TBBCs from Passage 3 to 4 and MSC from Passage 2 to 4 were harvested as routine using TrypLE, then 1:1 ($2 \times 10^6$ cells of each/injection) mixed with each other and resuspended in Dulbecco's Modified Eagle's Medium. Prior to subcutaneous injection, the cell suspension was further mixed 1:1 (v/v) with growth factor reduced Matrigel and then transplanted onto NSG mice with a total volume of 200 μl/injection. For the spheroid approach, MSCs were spun into cell pellets and pre-induced in chondrogenic differentiation medium for 72-h as described above. MSC spheroids ($10^6$ cells/spheroid) were submerged in TBBC suspension ($2 \times 10^6$ TBBC: 1 spheroid in a volume of 100 μl), which was pre-mixed 1:2 (v/v) with growth factor reduced Matrigel, to allow a formation of sandwich spheroids that covered by a shell of TBBCs. Once solidified, sandwich spheroids were subcutaneously transplanted onto NSG mice following minor surgery procedures. Animals were sacrificed at Week 4 post-transplantation and subcutaneous growths were collected for paraffin sectioning followed by histological examination. Male and female animals at 8 weeks old were used in experiments. All procedures were conducted under IACUC guidelines and approved protocol by the IACUC committee of Shanghai Sixth People's Hospital.

## Quantitative real-time PCR

Total RNA was extracted using a RNeasy Mini Kit (Qiagen) and treated with DNase I. Isolated RNA (1 μg) was subjected to reverse transcription using RevertAid First Strand cDNA Synthesis Kit (Thermo Scientific) according to the manufacturer's instructions.

Quantitative PCR analysis was carried out as follows: 10 ng of cDNA, 200 nM forward + reverse primers and 1× Power SYBR™ Green Master Mix (Applied Biosystems) in a total reaction volume of 20 μl, followed by amplification using a pre-installed protocol on the QuantStudio 5 System (Applied Biosystems). Amplification of human GAPDH was used as an endogenous control to standardize the amount of sample added to the reaction. Primers used are as follows: ACAN 5′-GTCTCACTGCCCAACTACCC-3′, reverse 5′AAAGTCGAGGGTGTAGCGTG-3′; SOX9 forward 5′-ACCACCCGGATTACAAGTACCA-3′, reverse 5′-TTGAAGAT GGCGTTGGGGGAG-3′; BGLAP forward 5′-TCACACTCCTCGCCCTATTG-3′, reverse 5′-GGGTCTCTTCACTACCTCGC-3′; RUNX2 forward 5′-AGGCAGTT CCCAAGCATTTCATCC-3′, reverse 5′-TGGCAGGTAGGTGTGGTAGTGAG-3′.

## RNA-seq, ATAC-Seq, and data analysis

Total RNA (RIN ≥ 9.0) from TBBC clones and ALI structures were subjected to RNA-Seq, and genomic DNA from TBBC clones was subjected to ATAC-Seq. Samples were sequenced and analyzed with assistance from Shanghai Jiayin Biotechnology Co., Ltd.

Total RNA (3 μg/sample) was used as input material for sample preparations. Sequencing libraries were generated using NEBNext Ultra RNA Library Prep Kit for Illumina (NEB, USA) following the manufacturer's recommendations and index codes were added to attribute sequences to each sample. Briefly, mRNA was purified from total RNA using poly-T oligo-attached magnetic beads and fragmented using divalent cations under elevated temperature in NEBNext First Strand Synthesis Reaction Buffer. First-strand cDNA was synthesized using random hexamer primer and M-MuLV Reverse Transcriptase (RNase H). Second strand cDNA synthesis was subsequently performed using DNA

Polymerase I and RNase H. Remaining overhangs were converted into blunt ends via exonuclease/polymerase activities. Adapter-ligated cDNA fragments (preferentially 150–200 bp in length) were selected using the AMPure XP system (Beckman Coulter, Beverly, USA), and then incubated with 3 μl USER Enzyme (NEB, USA) at 37 °C for 15 min, followed by 5 min at 95 °C. PCR was performed with Phusion High-Fidelity DNA polymerase, Universal PCR primers and Index (X) Primer. At last, PCR products were purified (AMPure XP system) and library quality was assessed on the Agilent Bioanalyzer 2100 system. RNA-Seq libraries were sequenced on an Illumina Novaseq 6000 platform and 150 bp pair-end reads were generated.

Purified TBBCs were harvested from culture via differential trypsinization to ensure complete clearance of feeder contaminant. Cells (50,000/sample) were resuspended in cold PBS and proceeded to chromatin extraction, Tn5-mediated fragmentation, and adapter incorporation according to ATAC-Seq protocol[65]. Reduced-cycle amplification was carried out to minimize PCR bias. After samples were PCR-amplified using 1× NEBNext High-Fidelity PCR Master Mix (New England Biolabs, MA), subsequent libraries were purified with the MinElute PCR Purification Kit (Qiagen) and subjected to sequencing on Illumina Novaseq 6000 using PE150.

For bioinformatics analyses of RNA-Seq, raw sequence reads were initially processed using FastQC (http://www.bioinformatics.babraham.ac.uk/projects/fastqc/) for quality control. Quality-filtered reads were then mapped to the human genome (hg38) using STAR, and only the uniquely mapped reads were kept. Read counts were calculated using HTSeq v0.6.0 and output as FPKM upon normalization. Differentially expressed genes were identified using the DEGseq algorithm (fold change ≥2, $p < 0.05$). Heatmap plots were drawn by the R based on the differential gene analysis. GO annotation from NCBI (http://www.ncbi.nlm.nih.gov/), UniProt (http://www.uniprot.org/), and the Gene Ontology (http://www.geneontology.org/) were used to perform GO analysis. Fisher's exact test was applied to identify the significant GO categories and FDR was used to correct the p-values. GO and Pathway (KEGG) analyses were also performed using Gene Set Enrichment Analysis (GSEA) software.

For bioinformatics analyses of ATAC-Seq, raw sequence reads were initially processed using FastQC for quality control. Clean reads were obtained from the raw data by removing the adapter sequences and then aligned to the human genome (hg38) using BWA-MEM. The bam files generated by the unique mapped reads were proceeded to peak calling using MACS2 and an initial threshold q value of 0.05 was set as the cutoff. Reads distribution (from bigwig) across genes was analyzed by deepTools and presented as heatmaps. To demonstrate the similarity between replicates, overlapped peaks between samples were visualized by UpSetR, followed by Pearson's correlation analysis. To identify differential accessible peaks, peak files of each sample were firstly merged using Bedtools, then the counts of reads over the bed were determined for each sample using Bedtools multicov, and differential accessible peaks were finally assessed using DESeq2 (fold change ≥2, $p < 0.05$). Motif analysis of differential accessible peaks was performed using HOMER's findMotifsGenome.pl tool. Peaks were annotated by using HOMER annotatePeaks.pl and distribution was plotted as Pie-chart using R. To further study the functional implication of changes in chromatin accessibility, ATAC-Seq results proceeded to integration analysis with RNA-Seq. Fold change values of RNA-Seq and ATAC-Seq for each gene were displayed in a four-quadrant plot, and genes with changes in both gene expression and chromatin accessibility (absolute $\log_2 FC \geq 1$) were shortlisted followed by Pearson's correlation analysis. Further, the overlapped DEGs between ATAC-seq and RNA-seq proceeded to Gene Ontology and Pathway (KEGG) enrichment analysis. Fisher's exact test was applied to identify the significant categories and FDR was used to correct the p-values. Chromatin at loci of interest was depicted by IGV (Integrative Genome Viewer) 2.8.6 using reference genome hg38. IGV tracks from ATAC-seq data were group-autoscaled to enable comparison.

## Statistical analysis

Statistical analyses were performed using Microsoft Excel or GraphPad Prism 9. All results were expressed as means ± SD unless otherwise stated in Figure Legends. The significance of difference was assessed by a two-tailed Student's *t*-test, and values of $p < 0.05$ were considered statistically significant.

## Reporting summary

Further information on research design is available in the Nature Research Reporting Summary linked to this article.

## Data availability

The raw data files of RNA sequencing and ATAC sequencing have been deposited in the Gene Expression Omnibus GEO data repository under accession code GSE153390. All other data supporting the findings of this study are available within this article, the Supplementary Information files, and the Source Data file. A reporting summary linked to this article is available as a Supplementary Information file. Source data are provided with this paper.

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

## Acknowledgements
The authors wish to thank Anthony J. Turner and Natalia N. Nalivaeva from the University of Leeds for their critical reading of the manuscript and the helpful comments. We are grateful to the laboratories of Martien C. Stuart at East China University of Science & Technology, Binglai Chen at Shanghai Houchao Biotechnology, and the Department of Orthopedics, Shanghai Sixth People's Hospital for equipment support. We appreciate Jiayin Biotechnology Ltd. (Shanghai, China), for technical assistance on sequencing assays. This work was funded by the National Natural Science Foundation of China 81900013 (Y.H.), 81930001, 81870055 (T.R.), and 81900059 (S.S.).

## Author contributions
Y. Hong and T.R. designed the project. Y. Hong instructed experiments and wrote the paper. Y. Hong, S.S., Y.D., and Z.L. performed the experimental studies. T.R., Y.G., H.H., and Q.Z. executed bronchoscopy, clinical judgment, and patient sampling. Y. Han performed the histologic assessment. M.H. advised on bioinformatic analysis and assisted with data interpretation.

## Competing interests
The authors declare no competing interests.
