## [Peer Review File · Nature Communications]

Malfunction of Airway Basal Stem Cells plays a crucial role in Pathophysiology of Tracheobronchopathia OsteoplasticaREVIEWER COMMENTS

Reviewer #1 (Remarks to the Author):

In this manuscript, Hong et al describe the isolation of basal stem cells from tracheobronchopathia osteochondroplastica (TO) patients followed by downstream functional and RNAseq/ATACseq characterization. This disease is rare, with a presumed significant level of underreporting and this study provides insight into how basal cells might become pathologic. Intriguingly, the disease is characterized by ectopic cartilage and osseous nodules in the trachea and bronchus. This study suggests that pathologic basal cells are capable of inducing cartilaginous spheroids, but not full ossification. Finally, the authors suggest that small molecule inhibition of BMP signaling may be of potential therapeutic value, however they show no evidence that this would occur nor a mechanism that would explain BMP action.

Major comments:

The single key missing piece in this manuscript is identifying the mechanism of cartilage induction by TO basal cells. Without this, the manuscript is not suitable for publication in Nature Communications. As this is the key phenotype of the disease, the mechanism that TO basal cells induces cartilage needs to be demonstrated by genetic modification of the TO basal cells showing a change in their ability to induce cartilage. Furthermore, it should be demonstrated whether or not this effect is linked to BMP signaling.

Tadokoro et al demonstrated that the effect of exogenous Bmp4 on basal cells is reversible after removal of Bmp4. In TO, basal cells appear to be in a positive feedback loop expressing excessive BMPs, it would be key to remove noggin from TO ALIs after treatment and see if this loop is ended, or does the treatment effect reverse? If noggin treatment results in a stable "rescue" of TO basal cells, demonstration that these basal cells cannot induce cartilage would greatly improve this manuscript. ATACseq analysis of TO basal cells after noggin administration should be performed to demonstrate whether or not BMP signaling is upstream of the epigenetic changes found in TO cells.

What is the effect of Noggin added to ALI of TO basal cells that are beyond stage I? There is a discrepancy where earlier, in-vivo differentiation assays in S6C of stage I TO basal cells exhibited mostly normal differentiation, and it was only higher stage basal cells that failed to differentiate properly.

Minor comments:

Methods: Please include the method used for feeder free basal cell culture, as well as the number of days the ALI were cultured for.

Fig 1B: For clarity, please mark whether images in 1B are from TO or non-TO basal cells.

Fig 1: It is recommended, if feasible, to have more replicates of RNAseq, as n=2 greatly reduces the number of genes that can be differentially detected.

Fig 3B: Please label 3B with NM95/equivalent on the panel for clarity. Please confirm/also include in the figure legend that the red outlined boxes are the magnified regions in D/E (or sequential sections of that region)

Fig 3F: Please clarify whether 3F is quantified xenografts or combined with ALI results (Figure legend says xenografts, text says xenograft and ALI sections).

Fig 4: Please show and quantify the ectopic cartilaginous formations if possible.

Fig S4: For clarity, please change curved lines to straight lines in the graphs, so that you are not extrapolating data.

Fig S6C: For clarity, please reorder the panels to flow from normal like->dysplasia

Reviewer #2 (Remarks to the Author):

Comment to Authors

Summary: In this interesting study to investigate the pathophysiology of TO, the authors demonstrate that human-derived TO cells have altered cell morphology, proliferation and differentiation potential, and RNA expression patterns. They further correlated these alterations with increased sites of open chromatin.

Although thought-provoking and providing new perspectives in considering TO pathology, the data are largely observational and correlative. Pathways and genes identified through RNAseq, and that are interpreted as relevant to disease progression including heterotopic ossification, have not been validated. Similarly, correlations between TO and increased chromatin accessibility are speculative and not demonstrated as relevant to the disease.

Specific comments

1. In the Introduction (page 5) and again in the Discussion (page 13) the authors use the term 'imprinting' to describe their conclusions. Since imprinting has a very specific biological/genetic meaning that is not relevant here, it is best to avoid using the term in the context of this study.

2. Page 8: Cell differentiation was assessed by quantifying immunofluorescent-stained tissue sections and IHC-labeled xenographs. Alternative or additional methods should be used to provide more convincing quantitative data.

3. Co-culture assays with TO-TBBCs and bone marrow MSCs show that MSCs were induced to express higher levels of aggrecan and Sox9 (page 10, Fig. S4). These are important data to support the authors' conclusions and would be more appropriate within a main figure. However, the data in Fig S4D shows very weak staining, making the data less convincing.

4. The MSCs used in co-culture experiments are described (page 10) as being pre-induced. The Methods describes apparent pre-induction with chondrogenic or osteogenic media, however how the cells were treated for the assays is not clear in the text, and the rationale for using a pre-treatment is not provided.

5. In some cases, it is difficult to determine what cells (basal, differentiated, source) were used for specific experiments and to understand what information could reasonably be gained from the in vitro and cell implant assays used.

Example: it is unclear what cells (described as "ALI samples") were evaluated by RNAseq in Figure 4.

6. No references were cited to support the statement (page 12): ...a pre-active status was detected on a list of genes essential for skeletal development in TO-TBBCs, including SATB2, ODAPH, CSGALNACT1, COL5A2 and BMP3...

7. No references were cited to support the statement (page 13): While upregulated genes are mainly enriched in categories related to cilium biosynthesis and function as well as Smoothened signaling, which is important for airway epithelial development and executing synergistic effect on BMP suppression, a down-turn of genes related to cartilage-bone morphogenesis were observed in treated pedigrees compared to TO-ALI under routine differentiation.

No references were cited to support the statement (page 14): While upregulated genes are mainly enriched in categories related to cilium biosynthesis and function as well as Smoothened signaling, which is important for airway epithelial development and executing synergistic effect on BMP suppression...

Additional experiments could confirm some of the authors' conclusions from their data:

- Up-regulation of BMP signaling could be confirmed by IHC for pSmad1/5.

- Noggin treatment rescue studies could be complemented with LDN receptor kinase inhibitor.
- Pathway analysis can be verified by detection of specific gene targets or pathway signaling.

9. The Discussion is nicely and clearly written.

Additional comments:

10. In the sentence on page 9 ["This favors a hypothesis that TO-TBBCs and derived differentiated cells may elicit excessive chondrogenesis and/or ossification in submucosa by orienting the mesenchyme, rather than themselves contributing to bony/cartilaginous tissue formation."]. It is not clear what 'themselves' refer to - both stem cells and differentiated cells?

Reviewer #3 (Remarks to the Author):

In this manuscript the authors seek to characterize airway basal stem cells from subjects with Tracheobronchopathia Osteoplastica (TO), to investigate the hypothesis that dysfunction in this cell population interacts with other mesenchymal populations to trigger the formation of nodules and disease pathology. The authors not only conduct characterization of the basal cells from TO donors but they try to causally test that ability of these basal cells to give rise to disordered epithelia in vitro and in vivo as well as trigger mesenchymal dysfunction, through an elaborate series of assays. In general the investigation is well done and comprehensive. My biggest concern is the representative nature of the results shown. It appears only a couple of donors are used for most experiments, given the high heterogeneity in basal cell populations both within and across donors, I would like to see many of the experiments throughout repeated using at least 3 if not 4 to 5 donors' cells. Comments below.

Figure 1A,B,C - It is difficult to recognize what I am looking at in Figure 1 beyond the BF and p63 images. Are the CK5, Ki67, CK14, and Foxj1 images of entire clones? Maybe put the DAPI channel images in the supplement? Before making a broad statement about the uniformity of staining, I would like to see some quantification or at least low mag images showing staining positivity across a larger number of clones. Also what passage are these clones? Does the staining vary by passage?

Regarding the clonogenic frequency data in Figure 1D, how many donors is this data based on? It is very surprising that the clonogenic freq data is so different yet canonical basal cell marker expression is not changed. Does p63, krt5 staining change in non-TO cells with increasing passage, corresponding with the diminishing clonogenic freq? Also a statement is made that the clone size for TO basal cell colonies are smaller, but no quantification is given. Please provide. This should be based on multiple donors as well.

Regarding the basal RNA-seq cell data in Figure 1 and 2. This data is quite interesting, but it appears to be generated on only 2 donors per group? This is really too small of a sample size to trust the results. This experiment should be repeated using basal cells from all 6 TO and 8 non-TO donor cells. A list of all differentially expressed genes and associated statistics should be provided in a supplemental table.

Again regarding the morphological assessments of basal cells made in Figure 2C,D, to make sweeping generalizations about TO basal cells. This must be based on at least 4 or 5 of donors, quantified, and then statistically analyzed to test for differences.

The xenograft experiments look quite convincing, yet I have two concerns regarding the representative nature of the results presented.

1. How many human TO and non-TO donor basal cells were transplanted? How many times were each donors' cells transplanted. I would want to see these experiments on several donors, and repeated several times. Also what passage were the transplanted basal cells?
2. How did the authors select the human cell transplanted areas to analyze? How many areas were analyzed? How consistent was the disordered epithelia observed in the TO transplanted mice?

The ALI results are intriguing but again I would want to see the RNA-seq performed in at least 3-4 donor ALI cultures from both the TO and non-TO groups. Additionally, I would like to see histological cross-sections of the ALI cultures so the structure of the epithelium can be observed.

Regarding the correlation of ATAC-seq peaks and nearby genes with RNA-seq based differential expression in TO. Considering the number of ATAC-seq peaks, it is expected that there would be some overlap with the RNA-seq results. For the correlation in results to be more meaningful the authors should perform an enrichment analysis with statistical test.

Throughout the authors talk about TO being an inflammatory condition and that inflammation drives disease, however they really don't show any evidence of inflammation in their numerous experiments with genome-wide assays. For example with the goblet cell metaplasia shown repeatedly throughout, is there any evidence for Type 2 cytokines? IL1 inflammation? Please report any inflammation data to support the assertion.

Relatedly, no data is provided to explain the goblet cell metaplasia observed? any evidence for SPDEF involvement?

Reviewer #4 (Remarks to the Author):

My remarks as a non-expert in Airway Basal Stem Cells is focused on the analyses of ATAC-seq.

The circos plot in Fig 5 is a little confusing. It seems to be used to illustrate the similarity between replicates. The smaller radii of some samples distorts the signal. A better way to demonstrate similarity between replicates would be by pairwise comparisons or correlations between biological replicates and groups.

There are no issues with ATAC-seq raw data processing and peak calling.

Is Fig 5e a contiguous locus containing the 4 labeled genes? A genome scale for each locus should be included for clarity. Similarly, a Y axis scale is important, unless is it consistent for all loci and can be mentioned in the legend.

Figs 5f/g, normalized read counts are shown for genes. Is this gene expression or chromatin accessibility? The legend says "Bar-chart showing transcriptional trends."

Based on Tables S1/S2, which mention only TSSs, it appears that distal regulatory elements were only mentioned in passing in the authors' analyses (such as the BMP3 locus). This is despite the observation presented in Fig 5D that the vast majority of peaks are not at TSSs. Are there meaningful biological differences at distal regulatory elements that are similar or different from the TSS focused analyses presented in the figures?

Reviewer #1 (Remarks to the Author):

In this manuscript, Hong et al describe the isolation of basal stem cells from tracheobronchopathia osteochondroplastica (TO) patients followed by downstream functional and RNAseq/ATACseq characterization. This disease is rare, with a presumed significant level of underreporting and this study provides insight into how basal cells might become pathologic. Intriguingly, the disease is characterized by ectopic cartilage and osseous nodules in the trachea and bronchus. This study suggests that pathologic basal cells are capable of inducing cartilaginous spheroids, but not full ossification. Finally, the authors suggest that small molecule inhibition of BMP signaling may be of potential therapeutic value, however they show no evidence that this would occur nor a mechanism that would explain BMP action.

Major comments:

The single key missing piece in this manuscript is identifying the mechanism of cartilage induction by TO basal cells. Without this, the manuscript is not suitable for publication in Nature Communications. As this is the key phenotype of the disease, the mechanism that TO basal cells induces cartilage needs to be demonstrated by genetic modification of the TO basal cells showing a change in their ability to induce cartilage. Furthermore, it should be demonstrated whether or not this effect is linked to BMP signaling.

Please see responses above.

Tadokoro et al demonstrated that the effect of exogenous Bmp4 on basal cells is reversible after removal of Bmp4. In TO, basal cells appear to be in a positive feedback loop expressing excessive BMPs, it would be key to remove noggin from TO ALIs after treatment and see if this loop is ended, or does the treatment effect reverse? If noggin treatment results in a stable “rescue” of TO basal cells, demonstration that these basal cells cannot induce cartilage would greatly improve this manuscript. ATACseq analysis of TO basal cells after noggin administration should be performed to demonstrate whether or not BMP signaling is upstream of the epigenetic changes found in TO cells.

Response:

It’s a very good point. A means of stable “rescue” of TO basal cells is thought to be of great therapeutic significance, however, based on our current findings, the treatment effect is sufficient only when persistent brake is applied on cells rather than transient inhibition at one stage. By histological observation, Noggin-treated TO basal cells failed to differentiate into mucociliary epithelium as normal, when inhibitor was withdrawn during the ALI process. This was subsequently examined by RNA-Seq and whole-genome PCA is displayed as below. It’s clearly demonstrated that ALI upon Noggin-on-stem-cell-only treatment was still in close proximity to untreated control and stayed in the TO cluster, whilst ALI upon Noggin-on-both-stem-and-ALI treatment shifted dramatically towards the non-TO cluster in the map and appeared far apart from untreated control.

Despite ATAC-seq analysis of TO basal cells after noggin administration yet to be done due to budget allocation, results so far suggest that TO basal cells retain autonomous positive feedback on BMP axis, which is more like an “effector” but less likely, at least by alone, to be the upstream of TO-associated epigenetic changes.

In comparison, the exogenous BMP treatment is likely to bring in only transient changes at transcriptional level, thus phenotypic alteration could be readily reserved once extracellular stimuli was removed.

Although BMP inhibition may not be a cure strategy for TO, through this work, we understand the pathogenesis of this “100-year-old mystery” disease much better than before and verified a participation of airway stem cell, which retains a memory of osteochondro-inductive signals, in this event. We have several speculations about upstream targets, of which chronic inflammation may be a candidate. However, whether these TO cells can be easily stably rescued or cell replacement would be a more applicable approach needs to be explored in follow-up studies.

What is the effect of Noggin added to ALI of TO basal cells that are beyond stage I? There is a discrepancy where earlier, in-vivo differentiation assays in S6C of stage I TO basal cells exhibited mostly normal differentiation, and it was only higher stage basal cells that failed to differentiate properly.

Response:

We appreciate the reviewer for the concern. Please be clarified that the discrepancy shown in supplementary Fig. S6 (Figure 6 in the revised manuscript) was about diverse differentiation potentials of basal cells collected from TO patients but derived from their nearby normal-looking tissues rather than lesion regions. This is a step-further study of disease progression aiming to explore whether there is early change(s) of cell potential occurred on surrounding epithelia prior to pathological changes. These patient-matched normal region derived basal cells (labelled with prefix of “PM”) were only discussed in the last two figures as extension. The main findings of stem cell malfunctions shown in Figure 1-5 were based on comparisons between non-TO controls and TO basal cells from patients’ absolute lesion regions. Impaired mucociliary differentiation capability is a consistent phenotype found on TO basal cells regardless of disease stage. In fact, the results shown in Fig. 6B (Figure 4f in the revised manuscript) was about the effect of Noggin on ALI of TO-02 (a stage III case) basal cells. Furthermore, we have also examined Noggin effect on multiple cases, and similar reverse effect on ciliogenesis could be detected. Please find below the results from TO-06 (a stage II case).

Minor comments:

Methods: Please include the method used for feeder free basal cell culture, as well as the number of days the ALI were cultured for.

Response:

Yes, information has been included in the revised manuscript.

Fig 1B: For clarity, please mark whether images in 1B are from TO or non-TO basal cells.

Response:

The original Fig.1B showed images of non-TO clones as examples because both non-To and TO cells are positive for p63/CK5/CK14/Ki67 and negative for CK10/Foxj1, despite different degrees of immunoreactivity against CK14 and Ki67 between non-TO and TO groups. In current Figure 1, images from both non-TO and TO clones have been included and displayed in parallel.

Fig 1: It is recommended, if feasible, to have more replicates of RNAseq, as n=2 greatly reduces the number of genes that can be differentially detected.

Response:

More replicates of RNAseq have been included, giving a sample volume n=6-7/group.

Fig 3B: Please label 3B with NM95/equivalent on the panel for clarity. Please confirm/also include in the figure legend that the red outlined boxes are the magnified regions in D/E (or sequential sections of that region)

Response:

Yes, 3B images have been labelled with "Human nucleoli", and the annotation of red outlined boxes has been added into figure legend.

Fig 3F: Please clarify whether 3F is quantified xenografts or combined with ALI results (Figure legend says xenografts, text says xenograft and ALI sections).

Response:

Fig. 3F was quantified xenografts combined with ALI results and Fig. 3G was quantified xenografts based on section images captured from random fields and by manual data collection. With the aid of HALO software, we re-quantified all three cell types using entire xenograft sections. Results have been included in current Figure 3f.

Fig 4: Please show and quantify the ectopic cartilaginous formations if possible.

Response:

There was no ectopic cartilaginous formation arisen on ALIs.

Fig S4: For clarity, please change curved lines to straight lines in the graphs, so that you are not extrapolating data.

Response:

Yes, graphs have been changed as suggested, and currently displayed in Figure 5b.

Fig S6C: For clarity, please reorder the panels to flow from normal like->dysplasia

Response:

The panels have been reordered, and which are currently shown in Figure 6.

Reviewer #2 (Remarks to the Author):

Comment to Authors

Summary: In this interesting study to investigate the pathophysiology of TO, the authors demonstrate that human-derived TO cells have altered cell morphology, proliferation and differentiation potential, and RNA expression patterns. They further correlated these alterations with increased sites of open chromatin.

Although thought-provoking and providing new perspectives in considering TO pathology, the data are largely observational and correlative. Pathways and genes identified through RNAseq, and that are interpreted as relevant to disease progression including heterotopic ossification, have not been validated. Similarly, correlations between TO and increased chromatin accessibility are speculative and not demonstrated as relevant to the disease.

Specific comments

1. In the Introduction (page 5) and again in the Discussion (page 13) the authors use the term ‘imprinting’ to describe their conclusions. Since imprinting has a very specific biological/genetic meaning that is not relevant here, it is best to avoid using the term in the context of this study.

Response:

To use “imprinting” in the context was denoting that there is a memory of pathogenic changes retained in TO basal cells, because they show intrinsic changes even after

isolation from original disease microenvironment and upon *in vitro* expansion. The term has been replaced in current description.

2. Page 8: Cell differentiation was assessed by quantifying immunofluorescent-stained tissue sections and IHC-labeled xenographs. Alternative or additional methods should be used to provide more convincing quantitative data.

Response:

With the aid of HALO image analysis software v3.0 coupled with NanoZoomer digital slide scanner, cell differentiation has been assessed by quantifying the entire IHC-labeled sections, with sample replication of 2-4 sections/xenograft/marker, 1-3 independent xenografts/sample and ≥ 6 samples/group. Data are presented as percentage of total transplanted (human nucleoli+) cells.

3. Co-culture assays with TO-TBBCs and bone marrow MSCs show that MSCs were induced to express higher levels of aggrecan and Sox9 (page 10, Fig. S4). These are important data to support the authors' conclusions and would be more appropriate within a main figure. However, the data in Fig S4D shows very weak staining, making the data less convincing.

Response:

Much appreciated for the reviewer's suggestion. Data of original Fig. S4 has been summarized into current Figure 5, in which additional data from *in vivo* co-culture assays were also included. HE staining of MSC spheroid sections has been displayed for better visualization of cell morphological change, and Alcian blue staining on fully differentiated MSC has been carried out in parallel as a positive control.

4. The MSCs used in co-culture experiments are described (page 10) as being pre-induced. The Methods describes apparent pre-induction with chondrogenic or osteogenic media, however how the cells were treated for the assays is not clear in the text, and the rationale for using a pre-treatment is not provided.

Response:

Details about co-culture assays have been added into Method.

"Pre-induced" MSC was chosen to perform co-culture assays, because under such condition, cells have been pre-set to lineage differentiation (For example, MSCs have been spun into pellets and cultured in chondrogenic differentiation medium for 72h to allow the formation of aggregated spheroids) but still at a state from which they could not continue with spontaneous differentiation upon a switch to basic medium, which excluded conventional inducers (please refer to results of pilot experiment below).

5. In some cases, it is difficult to determine what cells (basal, differentiated, source) were used for specific experiments and to understand what information could reasonably be gained from the in vitro and cell implant assays used.

Example: it is unclear what cells (described as “ALI samples”) were evaluated by RNAseq in Figure 4.

Response:

Thanks for the comment. Detailed information has been added into current text.

Briefly speaking, stem cell state = basal cells, both ALI and xenograft = differentiated cells including main functional cell types derived from basal cells and a percentage of self-renewing basal cells. Using ALI samples is beneficial to collect “pure” epithelial-specific information such as RNA-Seq here, in contrast, cell implant assays help to understand cells’ spontaneous commitment and potential interaction with other cells such as stromal cells in this study.

6. No references were cited to support the statement (page 12): ...a pre-active status was detected on a list of genes essential for skeletal development in TO-TBBCs, including SATB2, ODAPH, CSGALNACT1, COL5A2 and BMP3...

Response:

References have been added.

7. No references were cited to support the statement (page 13): While upregulated genes are mainly enriched in categories related to cilium biosynthesis and function as well as Smoothened signaling, which is important for airway epithelial development and executing synergistic effect on BMP suppression, a down-turn of genes related to cartilage-bone morphogenesis were observed in treated pedigrees compared to TO-ALI under routine differentiation.

Response:

Reference has been added.

No references were cited to support the statement (page 14): While upregulated genes are mainly enriched in categories related to cilium biosynthesis and function as well as Smoothened signaling, which is important for airway epithelial development and

executing synergistic effect on BMP suppression...

Response:

Same as above.

8. Additional experiments could confirm some of the authors' conclusions from their data:

- Up-regulation of BMP signaling could be confirmed by IHC for pSmad1/5.
- Noggin treatment rescue studies could be complemented with LDN receptor kinase inhibitor.
- Pathway analysis can be verified by detection of specific gene targets or pathway signaling.

Response:

We thank the reviewer for all constructive suggestions. Results about pSmad1/5 staining and Noggin/LDN treatments have been added into current manuscript. By GSEA analysis of RNA-Seq data and HOMER motif analysis of ATAC-Seq, clues of upregulated BMP signaling could be demonstrated consistently. BMP target genes such as Id3, KLF10, DLX2, SOX9 were enriched in TO ALIs. Transcriptional activation of JUNB, an immediate early target induced by BMP2, was detected by ATAC-seq, reflected on the increment of percentage of JUNB-responsible TF motifs in TO.

9. The Discussion is nicely and clearly written.

Response:

Thanks for the comment.

Additional comments:

10. In the sentence on page 9 ["This favors a hypothesis that TO-TBBCs and derived differentiated cells may elicit excessive chondrogenesis and/or ossification in submucosa by orienting the mesenchyme, rather than themselves contributing to bony/cartilaginous tissue formation."]. It is not clear what 'themselves' refer to - both stem cells and differentiated cells?

Response:

The term of "themselves" refers to stem cell TBBCs.

Reviewer #3 (Remarks to the Author):

In this manuscript the authors seek to characterize airway basal stem cells from subjects with Tracheobronchopathia Osteoplastica (TO), to investigate the hypothesis that dysfunction in this cell population interacts with other mesenchymal populations to trigger the formation of nodules and disease pathology. The authors not only conduct characterization of the basal cells from TO donors but they try to causally test that ability of these basal cells to give rise to disordered epithelia in vitro and in vivo as well as trigger mesenchymal dysfunction, through and elaborate series of assays. In general the investigation is well done and comprehensive. My biggest concern is the

representative nature of the results shown. It appears only a couple of donors are used for most experiments, given the high heterogeneity in basal cell populations both within and across donors, I would like to see many of the experiments throughout repeated using at least 3 if not 4 to 5 donors' cells.

Comments below.

Figure 1A,B,C - It is difficult to recognize what I am looking at in Figure 1 beyond the BF and p63 images. Are the CK5, Ki67, CK14, and Foxj1 images of entire clones? Maybe put the DAPI channel images in the supplement? Before making a broad statement about the uniformity of staining, I would like to see some quantification or at least low mag images showing staining positivity across a larger number of clones. Also what passage are these clones? Does the staining vary by passage?

Response:

The marker staining pictures have been re-organized and displayed in current Figure 1 and supplementary Figure 2. Low mag images are included in supplement to show wide-field results. Clones from P4 and P7 culture were subjected to immunostaining with similar marker panels. Consistent reactivities were detected between passages.

Regarding the clonogenic frequency data in Figure 1D, how many donors is this data based on? It is very surprising that the clonogenic freq data is so different yet canonical basal cell marker expression is not changed. Does p63, krt5 staining change in non-TO cells with increasing passage, corresponding with the diminishing clonogenic freq? Also a statement is made that the clone size for TO basal cell colonies are smaller, but no quantification is given. Please provide. This should be based on multiple donors as well.

Response:

The clonogenicity data is based on all donors. Despite varying decline rates across donors (for example, the rates of TO-1, TO-2 and TO-5 sharply dropped to 5% or below since P5, whilst in TO-03, 04 and 06 donors, the rates of 5% or below appeared since P8), the same trend of change was demonstrated. The smaller colonies usually formed by TO basal cells were evident in original Fig. 1c, current Figure 1b and supplementary Figure 2b. Immunoreactivity of p63 and Krt5 does not change with increasing passage in current culture system, this could be explained by an observation that only p63+/K5+ cells form colonies when passage, whilst negative cells do not have growth advantage and will be spontaneously removed from the culture.

Regarding the basal RNA-seq cell data in Figure 1 and 2. This data is quite interesting, but it appears to be generated on only 2 donors per group? This is really too small of a sample size to trust the results. This experiment should be repeated using basal cells from all 6 TO and 8 non-TO donor cells. A list of all differentially expressed genes and associated statistics should be provided in a supplemental table.

Response:

Sample size of basal cell RNA-Seq has been increased to 6-7 donors/group, and resulted data has been included in current manuscript.

Again regarding the morphological assessments of basal cells made in Figure 2C, D, to make sweeping generalizations about TO basal cells. This must be based on at least 4 or 5 of donors, quantified, and then statistically analyzed to test for differences.

Response:

The morphological change of TO basal cells from classic polygon to elongated fibroblast-like shape is frequently observed when culture under feeder-free environment. Quantification result has been included (please refer to supplementary Figure 2c).

Based on expanded knowledge gained from additional RNA-Seq and cell characterization studies, we found that NGFR has a dynamic heterogeneity in p63+ population, and double positive cells more uniformly appeared in large robust colonies. To be consistent, there is a trend of decreased NGFR expression observed in TO culture (supplementary Figure 2 and 1.6-fold down-regulation at RNA levels across all donors) despite a non-significant p value of 0.0589 based on RNA-Seq data.

The xenograft experiments look quite convincing, yet I have two concerns regarding the representative nature of the results presented.

1. How many human TO and non-TO donor basal cells were transplanted? How many times were each donors' cells transplanted. I would want to see these experiments on several donors, and repeated several times. Also what passage were the transplanted basal cells?

Response:

In vivo differentiation assay was performed as routine on every TO and non-TO case. At least one growth was ensured for each case and 3 repeats for TO donors which usually formed short epithelia or small transplanted nests. Transplantation repeats were performed on independent animals, and each animal carried two spots with one non-TO, one TO in parallel. All cells used in transplantation were harvested from P3-P4 culture.

2. How did the authors select the human cell transplanted areas to analyze? How many areas were analyzed? How consistent was the disordered epithelia observed in the TO transplanted mice?

Response:

The harvested nodules were proceeded with paraffin embedding and followed by serial sectioning. The cross-sections with certain intervals and maximum epithelial surfaces (identified based on human-nucleoli staining) were subjected to marker staining and the entire labelled sections were analyzed by HALO software. The situation of disordered epithelia is like example images shown below. Consistency is generally high across repeats and within a nodule.

The ALI results are intriguing but again I would want to see the RNA-seq performed in at least 3-4 donor ALI cultures from both the TO and non-TO groups. Additionally, I would like to see histological cross-sections of the ALI cultures so the structure of the epithelium can be observed.

Response:

Sample size of ALI RNA-Seq has been increased to 6-7 donors/group. The histological cross-sections of ALI cultures are included in current manuscript as suggested.

Regarding the correlation of ATAC-seq peaks and nearby genes with RNA-seq based differential expression in TO. Considering the number of ATAC-seq peaks, it is expected that there would be some overlap with the RNA-seq results. For the correlation in results to be more meaningful the authors should perform an enrichment analysis with statistical test.

Response:

A four-quadrant graph is included in current manuscript (Figure 7d) to show overlap results, and accompanied with enrichment analysis.

Throughout the authors talk about TO being an inflammatory condition and that inflammation drives disease, however they really don't show any evidence of inflammation in their numerous experiments with genome-wide assays. For example, with the goblet cell metaplasia shown repeatedly throughout, is there any evidence for Type 2 cytokines? IL1 inflammation? Please report any inflammation data to support the assertion.

Response:

Inflammation data has been included and discussed in current manuscript.

Relatedly, no data is provided to explain the goblet cell metaplasia observed? any evidence for SPDEF involvement?

Response:

In addition to the inflammation-related pathways enriched in TO according to RNA-seq data, a significant enrichment of SPDEF-binding motifs in TO open chromatin is thought to be another evidence supporting the occurrence of goblet cell metaplasia.

Reviewer #4 (Remarks to the Author):

My remarks as a non-expert in Airway Basal Stem Cells is focused on the analyses of ATAC-seq.

The circos plot in Fig 5 is a little confusing. It seems to be used to illustrate the similarity between replicates. The smaller radii of some samples distorts the signal. A better way to demonstrate similarity between replicates would be by pairwise comparisons or correlations between biological replicates and groups.

Response:

The circos plot has been replaced with correlation results, displaying in supplementary Figure 13. Please see more discussion in corresponding paragraph of the main text.

There are no issues with ATAC-seq raw data processing and peak calling.

Is Fig 5e a contiguous locus containing the 4 labeled genes? A genome scale for each locus should be included for clarity. Similarly, a Y axis scale is important, unless is it consistent for all loci and can be mentioned in the legend.

Response:

Data has been re-plotted using IGV with genome scale. Y axis scale is consistent for all loci and stated in figure legend.

Figs 5f/g, normalized read counts are shown for genes. Is this gene expression or chromatin accessibility? The legend says "Bar-chart showing transcriptional trends."

Response:

This is gene expression.

Based on Tables S1/S2, which mention only TSSs, it appears that distal regulatory elements were only mentioned in passing in the authors' analyses (such as the BMP3 locus). This is despite the observation presented in Fig 5D that the vast majority of peaks are not at TSSs. Are there meaningful biological differences at distal regulatory elements that are similar or different from the TSS focused analyses presented in the figures?

Response:

Similar to the way shown by supplementary table 1 and table 2, differential peaks occur in exon and intron regions and those associated with significant gene expression changes have been listed and summarized in current supplementary tables 2-6. How exactly these chromatin accessibility changes in distal elements correlate with and regulate the transcription of corresponding genes is to be validated by combinatorial approaches in future studies.

Based on RNA-ATCT analysis results here, meaningful biological differences could be detected in all promoter, exon and intron regions, but more enriched in those changed at promoter. One reason for the vast majority of peaks in intron region, is signal repetition. There are usually multiple peaks observed in a locus, with the same trend of change or sometimes opposite changes. Furthermore, we realized that biological differences predicted by differential peaks in intron are usually not "intron-only" but

also detectable in promoter or sometimes exon results. Thus, analysis focusing on promoter region seems to be adequate for initial analysis which covers the majority of suggestive information. With particular focus, more efforts will surely be needed to validate and further explore the distal regulatory elements.

REVIEWERS' COMMENTS

Reviewer #1 (Remarks to the Author):

Accept conditionally based on the below:

The primary concern at initial submission was the identification of the mechanism through which TO basal cells induced cartilage. A key experiment in this was the transient blockage of BMP signaling to see whether breaking the putative positive feedback loop of BMP would be sufficient to revert TO basal cells. The authors respond in the comments that they attempted this experiment and found that transient blocking of BMP signaling through Noggin was insufficient to reverse TO in ALI conditions both by histology of ALI as well as n=1 RNAseq analysis. (Noggin treatment throughout differentiation was capable of reverting the ALIs).

This data, while negative, demonstrates that the increase in BMP signaling is a key effector of the TO phenotype, and deserves to be included in the manuscript. An immunohistochemical comparison of Noggin throughout ALI differentiation and Noggin withdrawal would be sufficient.

The ATAC analysis of motifs enriched in TO basal cells could be strengthened by correlating whether or not the transcription factors associated with the motif is actually expressed or even increased in TO basal cells (Simply through existing RNAseq data)

Despite not directly addressing the primary concern in the revision, the authors have uncovered that BMP signaling is the effector for TO, significantly revised the text to not over-reach on the claim of "memory" or imprinting, and substantially increased the statistical rigor of their data.

Minor comments:

Fig2A: Please add a white spacer between the cross section of A-tub/muc5AC and the E-cad panels to that they are distinctly separate.

Page 12: The authors state that there are small cartilaginous islets that form in the surrounding mouse cells after xenografting human TO basal cells. If possible, please include this interesting data.

Fig 5C: Please separate/add more whitespace to indicate more separation between the TO/non-TO induction with the full induction control panel.

Page 19, regarding Fig5D/F: Please work to clarify the text as it is slightly confusing.

Fig 5F: The top image in this panel needs to be labeled as to whether it came from a TO or non-TO xenograft.

Reviewer #2 (Remarks to the Author):

The authors have revised the manuscript extensively in response to review and the manuscript is greatly improved. Much of the manuscript Results section has been fully re-written and presents the authors ideas more clearly.

The authors' investigation of the role of TBBCs in the development of chondrogenic nodules in TO, leads them to conclude that TO-basal cells may act to stimulate submucosal cartilage nodule formation through production of BMP/TGFb ligands that in turn induce cells within the tissue

environment toward chondro/osteogenic differentiation.

This is an interesting study that advances the understanding of TO pathology, however while providing a foundation for future investigations, little mechanistic insight to prove the authors' hypothesis is provided and the conclusions remain speculative.

Additional specific comments

The Results section describing Figure 5 data would benefit from revision. As examples:

The schematic does not provide a useful illustration of the experiment performed. The experimental design in Figure 5a needs to be explained in the figure legend; the schematic is not self-explanatory and the description in the Results section is not sufficiently informative. In Results describing Figure 5, 'pre-induced' should be explained as cultured in chondrogenic media and 'basic' media as DMEM without chondrogenic induction.

The results for Figure 5 data should be clear that the MSCs used in the experiments are wild-type MSCs.

Figure 6a is described as an in vivo differentiation assay (Results), however appears to be a histological analysis of patient samples (figure and legend).

Reviewer #3 (Remarks to the Author):

I believe the authors have done an excellent job addressing my concerns, and that this manuscript will be a significant addition to pathobiological understanding of TO, as well as understanding of basal cell dysfunction. No further comments.

REVIEWERS' COMMENTS

Reviewer #1 (Remarks to the Author):

Accept conditionally based on the below:

The primary concern at initial submission was the identification of the mechanism through which TO basal cells induced cartilage. A key experiment in this was the transient blockage of BMP signaling to see whether breaking the putative positive feedback loop of BMP would be sufficient to revert TO basal cells. The authors respond in the comments that they attempted this experiment and found that transient blocking of BMP signaling through Noggin was insufficient to reverse TO in ALI conditions both by histology of ALI as well as n=1 RNAseq analysis. (Noggin treatment throughout differentiation was capable of reverting the ALIs).

This data, while negative, demonstrates that the increase in BMP signaling is a key effector of the TO phenotype, and deserves to be included in the manuscript. **An immunohistochemical comparison of Noggin throughout ALI differentiation and Noggin withdrawal would be sufficient.**

Response:

An immunohistochemical comparison has been included, displaying in current supplementary Fig. 9b.

The ATAC analysis of motifs enriched in TO basal cells could be strengthened by correlating **whether or not the transcription factors associated with the motif is actually expressed or even increased in TO basal cells** (Simply through existing RNAseq data)

Response:

Based on RNA-seq data, motif enrichment has been correlated to expression of corresponding TFs in TO basal cells. Results has been shown in supplementary Fig. 15b.

Despite not directly addressing the primary concern in the revision, the authors have uncovered that BMP signaling is the effector for TO, significantly revised the text to not over-reach on the claim of “memory” or imprinting, and substantially increased the statistical rigor of their data.

Minor comments:

Fig2A: Please add a **white spacer** between the cross section of A-tub/muc5AC and the E-cad panels to that they are distinctly separate.

Response:

White spacer has been added to better separate the panels.

Page 12: The authors state that there are **small cartilaginous islets** that form in the surrounding mouse cells after xenografting human TO basal cells. If possible, please include this interesting data.

Response:

The observation of suspectable cartilaginous islets provided us the first hint to hypothesize that TO basal cells may have a function in chondro-osteo direction. However, the spontaneous islets in mouse portion were sporadic among TO xenografts and primarily based on cell morphological assessment. Thus, we hadn't counted them in formal figures previously and employed co-culture assays to help with investigation. Images showing a representative sample have been shown in current supplementary Fig. 5.

Fig 5C: Please **separate/add more whitespace** to indicate more separation between the TO/non-TO induction with the full induction control panel.

Response:

Images have been better separated.

Page 19, regarding Fig5D/F: Please work to **clarify the text** as it is slightly confusing.

Response:

Corresponding text has been modified.

Fig 5F: The top image in this **panel needs to be labeled** as to whether it came from a TO or non-TO xenograft.

Response:

The panel has been rearranged with clearer labels.

Reviewer #2 (Remarks to the Author):

The authors have revised the manuscript extensively in response to review and the manuscript is greatly improved. Much of the manuscript Results section has been fully re-written and presents the authors ideas more clearly.

The authors' investigation of the role of TBBCs in the development of chondrogenic nodules in TO, leads them to conclude that TO-basal cells may act to stimulate submucosal cartilage nodule formation through production of BMP/TGFb ligands that in turn induce cells within the tissue environment toward chondro/osteogenic differentiation.

This is an interesting study that advances the understanding of TO pathology, however while providing a foundation for future investigations, little mechanistic insight to prove the authors' hypothesis is provided and the conclusions remain speculative.

Additional specific comments

The Results section describing Figure 5 data would benefit from revision. As examples:

The schematic does not provide a useful illustration of the experiment performed. The **experimental design in Figure 5a needs to be explained in the figure legend**; the schematic is not self-explanatory and the **description in the Results section** is not sufficiently informative.

Response:

Figure 5a has been modified for better illustration, and explained in figure legend.

In Results describing Figure 5, “pre-induced” should be explained as cultured in chondrogenic media and ‘basic’ media as DMEM without chondrogenic induction.

Response:

Information has been added into results.

The results for Figure 5 data should be clear that the MSCs used in the experiments are wild-type MSCs.

Response:

Information has been added into results.

Figure 6a is described as an in vivo differentiation assay (Results), however appears to be a histological analysis of patient samples (figure and legend).

Response:

Both results and figure legends sections have been checked and confirmed to be consistent.

Reviewer #3 (Remarks to the Author):

I believe the authors have done an excellent job addressing my concerns, and that this manuscript will be a significant addition to pathobiological understanding of TO, as well as understanding of basal cell dysfunction. No further comments.